# Interred mechanisms of resistance and host immune evasion revealed through network-connectivity analysis of *M. tuberculosis* complex graph pangenome

**Monica E. Espinoza,**[1] **Ashley M. Swing,**[1,2] **Afif Elghraoui,**[1,3,4] **Samuel J. Modlin,**[1] **Faramarz Valafar**[1]

**ABSTRACT**  *Mycobacterium tuberculosis* complex successfully adapts to environmental pressures through mechanisms of rapid adaptation which remain poorly understood despite knowledge gained through decades of research. In this study, we used 110 reference-quality, complete *de novo* assembled, long-read sequenced clinical genomes to study patterns of structural adaptation through a graph-based pangenome analysis, elucidating rarely studied mechanisms that enable enhanced clinical phenotypes offering a novel perspective to the species' adaptation. Across isolates, we identified a pangenome of 4,325 genes (3,767 core and 558 accessory), revealing 290 novel genes, and a substantially more complete account of difficult-to-sequence *esx/pe/pgrs/ppe* genes. Seventy-four percent of core genes were deemed non-essential *in vitro*, 38% of which support the pathogen's survival *in vivo*, suggesting a need to broaden current perspectives on essentiality. Through information-theoretic analysis, we reveal the *ppe* genes that contribute most to the species' diversity—several with known consequences for antigenic variation and immune evasion. Construction of a graph pangenome revealed topological variations that implicate genes known to modulate host immunity (*Rv0071-73, Rv2817c, cas2*), defense against phages/viruses (*cas2, csm6,* and *Rv2817c-2821c*), and others associated with host tissue colonization. Here, the prominent trehalose transport pathway stands out for its involvement in caseous granuloma catabolism and the development of post-primary disease. We show paralogous duplications of genes implicated in bedaquiline (*mmpL5* in all L1 isolates) and ethambutol (*embC-A*) resistance, with a paralogous duplication of its regulator (*embR*) in 96 isolates. We provide hypotheses for novel mechanisms of immune evasion and antibiotic resistance through gene dosing that can escape detection by molecular diagnostics.

**IMPORTANCE**  *M. tuberculosis* complex (MTBC) has killed over a billion people in the past 200 years alone and continues to kill nearly 1.5 million annually. The pathogen has a versatile ability to diversify under immune and drug pressure and survive, even becoming antibiotic persistent or resistant in the face of harsh chemotherapy. For proper diagnosis and design of an appropriate treatment regimen, a full understanding of this diversification and its clinical consequences is desperately needed. A mechanism of diversification that is rarely studied systematically is MTBC's ability to structurally change its genome. In this article, we have *de novo* assembled 110 clinical genomes (the largest *de novo* assembled set to date) and performed a pangenomic analysis. Our pangenome provides structural variation-based hypotheses for novel mechanisms of immune evasion and antibiotic resistance through gene dosing that can compromise molecular diagnostics and lead to further emergence of antibiotic resistance.

**KEYWORDS**  pangenome, tuberculosis, drug resistance, immune evasion, genetic diversity, network analysis, Shannon's entropy

Address correspondence to Faramarz Valafar, faramarz@sdsu.edu.

Monica E. Espinoza and Ashley M. Swing contributed equally to this article. Author order was determined both alphabetically and in order of increasing seniority.

The authors declare no conflict of interest.

See the funding table on p. 26.

*M*ycobacterium tuberculosis complex (MTBC) pathogens remain among the deadliest infectious agents in the world, with *Mycobacterium tuberculosis*, the causative agent of tuberculosis (TB) disease (1), alone killing an estimated 1.5 million people annually (2). With no effective vaccine (2–4), TB has cost the lives of over one billion people in the last 200 years (5).

MTBC strains exceptionally evade host immunity and resist antibiotic treatment. The range of phenotypic presentations (including immune evasion) by this pathogen cannot be explained by known single nucleotide polymorphisms (SNPs) alone, warranting more investigation of the full spectrum of genetic determinants of its diversity (6–8). While most projects seek SNPs to explain this phenotypic diversity, previous reports show that gene loss/gain can cause phenotypic change (9, 10). Gene gain mechanisms in prokaryotes generally rely on acquiring external genetic content through horizontal gene transfer, duplication (11), and gene recombination (12). While MTBC lacks horizontal acquisition of genetic material (12–14), it is hypothesized to have more genetic diversity than expected (15). Gene duplication and recombination have been observed in MTBC clinical strains (16, 17). Duplication events are known drivers of diversity in gene families and are subject to recombination with local homologous genes (18). These gain events inevitably result in several fates: maintenance of exact copies, partitioned function (subfunctionalization), new function (neofunctionalization), or loss of function (nonfunctionalization) (19).

Gene loss is also a known mechanism of diversity in MTBC (20) and encompasses frameshift mutations and structural variants (SVs) (both large and small) including deletions. Many studies have identified specific deletions, termed RvD1-6 and TbD1, in the most studied *M. tuberculosis* reference strain, H37Rv, with respect to *Mycobacterium* human-adapted lineages (21, 22). These large deletions and others (23) have been found in many clinical isolates. The reported differences between genomes are posited to be ancestrally acquired and have since been lost in some strains, as was the case with H37Rv (24, 25).

While a few studies have employed long-read sequencing (26–30), many have used short-read sequencing to assess the genomic diversity of *M. tuberculosis* through the construction of its pangenome. Short-read sequencing imposes limitations that might not be overcome post-sequencing (20, 26, 30–33). Short-read sequencing has a well-documented systematic bias that can introduce indels in some genomic regions (34) and does not capture sequence information in other genomic regions (35). As a result, it might be difficult to capture repetitive (35–37) or GC-rich (35, 38) regions (well-known genomic signatures of MTBC including the canonical and difficult-to-characterize *pe/ppe* families (35, 39)) and preclude complete *de novo* assembly, instead resulting in dozens to hundreds of contigs. This necessitates assumptions about the structure of the genome, using a reference, that may be unwarranted. Similarly, important events such as duplications can go unnoticed as short reads cannot always resolve the locus of homologous sequences (e.g., duplications) using their flanking regions. Alternative methods for processing these data, such as reference-based alignment, are similarly hindered (40), and previous work from our group has corrected reference-based calls in the past (35). SV detection using reference-aligned short reads has substantially lower sensitivity than long reads (41, 42). Overall, there is an opportunity for long-read sequenced, *de novo* assembled genomes to reveal distinct areas of genetic diversity yet unexplored in the MTBC pangenome.

Long-read sequencing enables the complete assembly of genomes *de novo* (without assumption of genomic structure) which can provide a complete assessment of genomic diversity both in terms of short and long SVs. *De novo* assembled MTBC genomes improve the detection of larger SVs, resolve the exact location of homologous regions including exact duplications, and, when the sequencing is properly performed (43), comprehensively identify insertions and deletions with minimal error. Studies of a limited number of *de novo* assembled MTBC genomes revealed such genomic changes (44–47). A few pangenome studies employing long-read sequencing of different constituents of

MTBC, specific lineages of *M. tuberculosis,* and causative agents for cutaneous tuberculosis have demonstrated the benefits of long-read sequencing for characterizing genetic diversity (26–30). Application of long-read sequencing to *M. tuberculosis* for distinct purposes separate from pangenome analyses has additionally been found, showing the promise of such technology to many fields of tuberculosis research (48–51). It should be considered that long-read sequencing can incur a high cost per base, misidentify single-base variants at a considerable rate, and reach low read depth if one is not careful. However, our study contributes to past genome comparison studies (26, 29, 52–57), which have employed some *de novo* assembled genomes, to accurately describe the diversity of MTBC in these difficult-to-characterize regions.

The foci of recent pangenome studies are diverse, spanning the discovery of vaccine candidates, drug targets, and antibiotic resistance targets (56–62) (Data Set S1). Many studies focus on the pangenomes of specific lineages, comparative genomics across infection sites (63, 64), or include non-tubercule or animal-adapted constituents of MTBC in pangenome calculations (27, 29, 65–77). Other pangenome studies focus solely on specific mechanisms of diversification (20, 26, 30, 32, 33). Pangenome-centered studies previously focused on characterizing pangenomes for specific lineages or assessed genetic diversity across *Mycobacteria* by including isolates from animal-associated *Mycobacteria,* all MTBC-forming bacteria*,* or non-*Mycobacteria* samples in pangenome calculations (29, 54, 66). These studies have also reported whether the shared genetic pool among *Mycobacteria* is open and expanding or closed. While pangenome studies on the shared genetic pool between human-associated and animal-associated pangenomes are important, the inclusion of non-tuberculous *Mycobacteria* strains has reported a smaller and more highly conserved core genome (since conservation is defined as genomic features shared among these varied-host strains). Focusing on these core genomes stands to wash out the clinically relevant genetic diversity and variation of MTBC strains in human infection. This variability could explain differences in virulence, viability in human tissues, and immune evasion, thus limiting efforts to reduce MTBC's global impact. Also, work reporting the pangenome of specific lineages can help identify new sub-lineages of MTBC and attenuate overestimations of transmission clusters. However, expanded emphasis on genetic variation across lineages of human-adapted MTBC strains could elucidate more global mechanisms of genetic diversity, subsequent evolution, and its contribution to factors such as conferral of drug resistance and persistence.

Our contribution to MTBC pangenome studies is our 109 long-read sequenced, *de novo* assembled *M. tuberculosis* complex genomes, the most reference-quality genomes of *M. tuberculosis* complex published with long-read sequencing. We conducted a pangenome analysis on these highly drug-resistant clinical isolates of *M. tuberculosis sensu stricto*, *M. tuberculosis* var. africanum plus H37Rv reference strain using the tool Panaroo. Panaroo shares information between input genomes using a graph-based method (78). It reports gene adjacency based on gene coordinates existing on the same contig, mending gene mis-annotations with this merged assembly information across genomes, thus improving the accuracy of the estimated core and accessory genomes in comparison to other pangenome construction methods (78). This study is one of the first pangenome network analyses applied to MTBC isolates. These findings not only characterize the different genomic structures and mechanisms of gene gain and loss of clinical MTBC isolates but also shed light on important regions of genomic diversity among genes engaged in host evasion and drug resistance and highlight gaps in our understanding of this pathogen and directions for future study.

## RESULTS

The pangenome of 109 MTBC clinical isolates and H37Rv contained a total of 4,325 genes, reflecting the total number of genes collectively present in any of the genomes. A vast majority (87%) of the genes were conserved across nearly all isolates, with 3,767 genes observed in at least 99% of the isolates (i.e., core genome). Conversely, 558 genes

(13%) were in the accessory genome, representing the genes that are shared by less than 99% of the genomes and reflect the structural genomic diversity of the *M. tuberculosis* complex (Data Set S2). This data set included lineage 1 (L1) isolates ($N = 20$), lineage 2 (L2) isolates ($N = 37$), lineage 3 (L3) isolates ($N = 10$), lineage 4 (L4) isolates ($N = 38$), lineage 5 (L5) isolates ($N = 5$), a lineage 6 (L6) isolate ($N = 1$), and a lineage 7 (L7) isolate ($N = 1$). There were 3 multi-drug-resistant isolates (MDR), 22 mono-resistant isolates, 3 pan-susceptible isolates, 10 pre-extensively (XDR) drug-resistant isolates, and 8 isolates with other drug resistance patterns. There were 11 isolates for which drug resistance information was not available (Table 1; Fig. 1). To understand the functional distribution of genes across core and accessory genomes, a GO enrichment analysis was performed powered by the PANTHER database. The core genome reports biological processes of DNA replication, transcription, and translation, but additionally, processes regarding lipid transport/localization, closely associated fatty-acid metabolism processes entangled with ATP synthesis and oxidative respiration, enoyl-coA hydratase activity, and glutamine family amino acid biosynthetic processes (Fig. S1) (79).

When stratifying by lineage-specific pangenomes, the total number of *ppe*, *pe_pgrs*, and *esx* family genes did not differ greatly and were similar in count and composition compared to other studies, though slightly attenuated for PGRS family genes (Table 1; Table S1). L4 core genome contained only 3,678 (87%) genes, representing the smallest core genome among the first five lineages (other lineages: 94%–98%). However, L4 also had the most lineage-specific genes (92 genes) not shared with other lineages (Fig. 1A; Data Set S2). Although there were more L4 isolates than other lineages in our sample, the diversity seen in our L4 pangenome is consistent with what has previously been noted

**TABLE 1** Global characteristics of the pangenome of mostly drug-resistant clinical *M. tuberculosis* complex strains by five most prevalent lineages

| Characteristic | All isolates | Indo-Oceanic (L1) | East Asian (L2) | East African-Indian (L3) | Euro-American (L4) | African 1 (L5) |
|---|---|---|---|---|---|---|
| Number of isolates in analysis, N (%) | 110 (100%) | 20 (18%) | 37 (33%) | 10 (9%) | 38 (34%) | 3 (3%) |
| Total pangenome/mean genome size | 4,325/4,022 | 4,133/4,017[b] | 4,105/4,027[b] | 4,099/4,041[b] | 4,209/4,016[b] | 4,051/4,023[b] |
| Genes in core genome | 3,767 (87%) | 3,767 (91%) | 3,767 (92%) | 3,767 (92%) | 3,767 (90%) | 3,767 (93%) |
| Genes in accessory genome | 558 (13%) | 366 (9%) | 338 (8%) | 332 (8%) | 442 (10%) | 284 (7%) |
| Genes in lineage core genome | NA[d] | 3,886 (94%) | 3,915 (95%) | 3,954 (97%) | 3,678 (87%) | 3,992 (98%) |
| Genes in lineage accessory genome | NA | 247 (6%) | 190 (5%) | 145 (3%) | 531 (13%) | 59 (2%) |
| Unnamed *ab initio* predicted genes | 290 (7%) | 222 (5%) | 205 (5%) | 200 (5%) | 219 (5%) | 191 (5%) |
| Essential genes,[a] N/total N (%) | 715/3,766 (20.3%) | 737/3,578 (20.6%) | 742/3,567 (20.8%) | 731/3,531 (20.7%) | 730/3,529 (20.7%) | 725/3,510 (20.7%) |
| Number of PPE genes,[c] median (min-max) | 68 (61–73) | 65 (61–68) | 71 (65–73) | 67.5 (62–69) | 68 (63–73) | 68 (68–68) |
| Number of PE genes,[c] median (min-max) | 33 (30–33) | 33 (30–33) | 33 (31–33) | 32 (30–32) | 32 (30–33) | 33 (33–33) |
| Number of PE_PGRS genes,[c] median (min-max) | 58 (54–63) | 55.5 (52–56) | 57 (54–61) | 58 (57–60) | 60 (55–63) | 59 (58–60) |
| Number of ESX genes,[c] median (min-max) | 23 (20–31) | 22 (20–24) | 23 (21–25) | 23 (21–23) | 24 (21–31) | 22 (22–22) |
| MDR | 3 | 1 | 0 | 1 | 1 | 0 |
| Mono-resistant | 22 | 11 | 2 | 1 | 7 | 0 |
| Other | 8 | 0 | 2 | 1 | 5 | 0 |
| Pan-susceptible | 3 | 0 | 0 | 0 | 3 | 0 |
| Pre-XDR | 10 | 3 | 3 | 0 | 4 | 0 |
| XDR | 53 | 2 | 30 | 4 | 17 | 0 |
| Missing | 11 | 3 | 0 | 3 | 1 | 3 |

[a]Essential defined as "ES," "ED," or "GA." Not essential defined as "GD," "NE," or "Uncertain" from DeJesus et al. (80). Denominators for essential genes represent lineage-specific pangenomes with genes removed for missing essentiality information.

[b]P-value < 0.001 using one-way Welch's ANOVA and Games-Howell test for multiple pairwise comparisons. Drug susceptibility testing was produced using previously described methods (81, 82).

[c]The number of genes is the sum of open reading frames for all locus tags identified by Panaroo as being a part of that gene family, including duplicate gene copies and merged annotations.

[d]NA, not applicable.

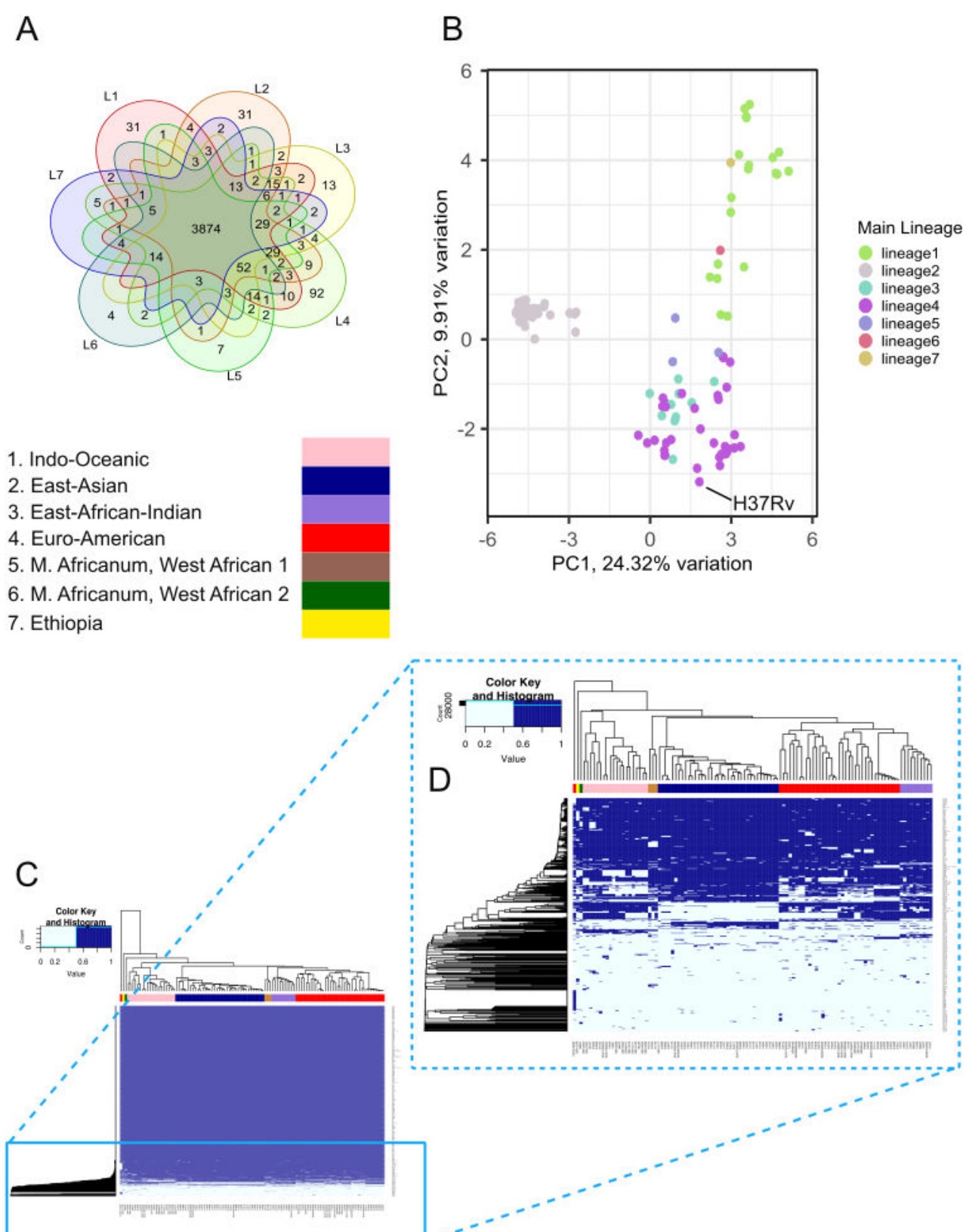

**FIG 1** Distribution of genes shared by lineage in the pangenome. (A) Diagram showing both the number of genes shared and unique to the lineages in the study sample (*N* = 110 isolates). Lineage-specific pangenome gene lists represent genes that are found in at least one isolate in the lineage. (B) Bi-plot of first two principal components, based on the presence/absence of genes in the pangenome, shown by lineage. The location of H37Rv on the bi-plot is indicated by the black line. (C) Heat map of full pangenome annotated by lineage. As most of the genes are conserved by all isolates in the pangenome, the distribution of genes in the accessory genome (D) is displayed in the outlet.

elsewhere (83, 84). It is important to note though that after removing H37Rv from the L4 lineage-specific pangenome analysis, the number of genes shared across all L4 isolates increased from 3,678 to 3,826 genes (91% shared).

Panaroo also identified 290 genes that were predicted *ab initio* as an open reading frame but not assigned a name by Prokka and labeled by Panaroo with a numeric "group" name (e.g., group_123). Approximately 96% (279/290) of these unnamed, *ab initio* predicted genes (hereby referred to as putative novel genes) were classified as hypothetical proteins by Prokka (Table S2). These putative novel genes consistently comprised about 5% of the lineage-specific pangenomes and nearly half of these genes were conserved in the overall core genome of all isolates ($N$ = 130, 44.8%; Table 1).

## Clustering and principal component analysis of gene presence profiles reveal putative novel genes to be significant distinguishing factors between lineages

Principal component analysis (PCA) and cluster analyses were collectively used to evaluate variation in genomic composition between isolates and assess whether the genomic structure of presence/absence clustered by lineage. We also sought to identify genes whose presence or absence may be driving gene-level differences between isolates. Results from the PCA bi-plot of the first two components illustrated a distinct clustering of L2 and L1 isolates, where L2 isolates more strongly correlated with the second principal component (Fig. 1B). By contrast, L3, L4, and L5 were less distinctly clustered.

The combination of genes accounting for the most variance in the first axis (principal component 1) was *Rv0071-73* and *Rv2817c-Rv2816c* (Pearson $r^2$ = 0.87–0.89, $P$-value < 0.0001; Fig. S2A). This is likely related to the fact that L2 isolates are known to have a deletion of the region of difference (RD) 105 that results in a gene fusion of *Rv0071* and *Rv0074*, and thus are a distinguishing factor in the PCA (85). Genes most significantly correlated with the second principal component were *tuf* (*Rv0685*), *ppe34* (*Rv1917c*), and putative novel genes group_73, group_326, and group_233 (Pearson $r^2$ = 0.45–0.49, $P$-value < 0.0001, Fig. S2A).

The genes that constituted the top 5% of the PCA loadings ($N$ = 67 genes) for the first five principal components were largely duplicated and putative novel genes ($N$ = 31/67 genes, 46%, Fig. S2B). Given their small numbers in the pangenome, we hypothesized that duplicated and putative novel genes were enriched in the top PCA loadings, and thus important for explaining the most variance between the components. To test for over-enrichment, we conducted a randomized permutation test without replacement to assess the baseline proportion of putative novel and duplicated genes we would expect to see if randomly sampling the same number of genes across the pangenome. In all the permutations ($N$ = 10,000), the expected proportion of putative novel genes randomly selected ranged from 0% to 19% and 0% to 14% for duplicates; there was not one randomized sample that matched or exceeded the proportion of putative novel or duplicated genes observed in our top 5% loadings (observed proportion = 23.88% for putative novel genes, 22.39% for duplicated genes) (Fig. S3). These findings highlight the presence of these putative novel and duplicated genes as influential in differentiating the principal components, and thus their presence may be important to distinguishing lineages (Fig. S3A and B).

An unsupervised hierarchical cluster analysis based on binary presence/absence of genes supported the findings of the principal component analysis and revealed most of the pangenome to be conserved among our isolates (Fig. 1C and D). Gene presence patterns largely clustered isolates by lineage, though H37Rv singularly did not cluster with any other lineage or isolate. This clustering pattern could reflect diverging evolution between H37Rv and the more recent clinical MTBC strains from the last common ancestor or could be influenced by the fact that H37Rv alone did not have any putative novel genes reported by Panaroo in its genome.

## Genes important to *in vivo* survival are conserved in the core, though many are regarded as "non-essential"

As we would expect genes with essential function to be conserved in all isolates, we assessed the distribution of essential genes across the core and accessory genomes using definitions derived from DeJesus et al. (80) and Zhang et al. (86) *in vitro* transposon mutagenesis (TnSeq) studies, as well as Bosch et al.'s CRISPR interfering (CRISPRi) study (87). Essential genes as defined by these studies reflected very similar distributions across the overall core and accessory genomes. Among the genes with no essentiality data missing from TnSeq studies, essential genes have over four times increased odds of being in the core compared to non-essential genes (DeJesus et al.: 4.41, 95% CI: 2.6–8.4, FDR corrected *P*-value = 2.79 × 10$^{-6}$; Zhang et al.: 4.26, 95% CI: 2.4–8.3, FDR-corrected *P*-value = 1.06 × 10$^{-5}$, Table 2). The percent of essential genes present in the lineage-specific pangenomes did not significantly differ by lineage (Table 1). Compared to CRISPRi experiments, odds ratios were slightly attenuated, with the odds of essential genes in the core being similar between H37Rv and a lineage 2 hypervirulent strain HN878 (H37Rv: 2.29, 95% CI: 1.61–3.37; HN878: 2.19, 95% CI: 1.54–3.23).

While the core was significantly enriched with essential genes, there were many conserved genes that were regarded as "not essential" by both transposon mutagenesis (*N* = 2,515, 74% core genome) and CRISPRi studies (*N* = 2,740, 82% core genome). These genes may not be essential *in vitro* but may provide important functions to bacterial survival *in vivo* under certain conditions. To this end, we assessed whether the core genome was enriched with known functions important to the survival, evolution, and pathogenicity of MTBC, many of which have been established as significant in *in vivo* animal models. Orthologous genes shared across *Mycobacterium leprae*, *Mycobacterium smegmatis*, *Mycobacterium avium,* and *Mycobacterium abscessus* have 6.64 times the odds of being conserved in the core, highlighting genes that are likely broadly essential to *Mycobacteriu*m species (FDR corrected *P*-value = 1.18 × 10$^{-25}$; Table 2). Of the 1,006 orthologous genes shared across these species that are present in our pangenome, 985 (97.9%) genes were recapitulated in our MTBC core genome. Other gene sets were also significantly enriched in the core genome, including genes known to limit drug efficacy, genes with MamA, MamB, or MamC DNA methylation motif sites in their promoters (88), genes whose transposon-inserted TA dinucleotide sites are underrepresented post-infection in mouse models (89), and transcription factors (TF). However, the statistical significance of core enrichment of genes with methylated promoters and TFs was lost after adjusting for the false discovery rate (FDR; Table 2). Conversely, *pe/ppe* genes had significantly reduced odds of being in the core compared to the accessory genome, highlighting their variability in number across our sample (Table 2). Nine hundred sixty-one genes from the gene sets significantly enriched in the core were labeled as "not essential" according to the *in vitro* DeJesus et al.'s study, accounting for approximately 38% (961/2515) of all the "not essential" genes in our core genome. These findings highlight genes determined as essential may be grossly underestimated from *in vitro* studies alone, as many genes' essentiality may come from functions that do not maximize growth rate or may prove essential in a certain particular environmental context.

## ESX, PE, PE_PGRS, and PPE gene families

MTBC genomes harbor four gene families that carry clinically important roles such as virulence regulation (90–92), host-pathogen interaction (93–96), and immune evasion (47, 93, 97, 98). Unfortunately, mainly due to sequence similarity, they present significant challenges in most Whole-Genome Sequencing projects and therefore are understudied. Our use of long-read sequencing allows us to overcome these challenges. The diversity in the presence and count of these genes can attenuate or enhance the aforementioned phenotypic characteristics of an isolate, making the correct identification and characterization of these gene families' constituents vastly important (99, 100).

**TABLE 2** Univariate and bivariate analyses of essential genes and gene function categories by core vs. accessory genome

| | No. genes | % pan-genome | Core N (%) | Access. N (%) | Crude OR (95% CI) | P-value | FDR-adjust. P-value |
|---|---|---|---|---|---|---|---|
| Essentiality studies | | | | | | | |
| DeJesus et al.[a] | | | | | | | |
| Essential gene | 450 | 12% | 439 (97.6%) | 11 (2.4%) | 4.41 (2.57–8.38)[e] | 6.85e−07[g] | 2.79e−06[g] |
| Essential domain | 26 | 1% | 25 (96.2%) | 1 (3.8%) | NA[i] | NA | NA |
| Growth advantage | 287 | 8% | 251 (87.5%) | 36 (12.5%) | 0.80 (0.56–1.17) | 0.226 | 0.329 |
| Growth defect | 130 | 3% | 124 (95.4%) | 6 (4.6%) | 2.36 (1.12–6.06) | 0.042[f] | 0.521 |
| Not essential | 2,802 | 74% | 2,515 (89.8%) | 286 (10.2%) | Ref | NA | NA |
| Uncertain | 71 | 2% | 54 (76.1%) | 17 (23.9%) | 0.36 (0.21–0.65) | 3.71e−04[g] | 9.89e−04[g] |
| Zhang et al.[b] | | | | | | | |
| Essential domain | 415 | 11% | 404 (97.3%) | 11 (2.7%) | 4.26 (2.43–8.34) | 3.32e−06[g] | 1.06e−05[g] |
| Not essential domain | 2,887 | 77% | 2,587 (89.6%) | 300 (10.4%) | Ref | NA | NA |
| Both regions | 294 | 8% | 269 (91.5%) | 25 (8.5%) | 1.25 (0.83–1.96) | 0.309 | 0.412 |
| Unable to determine | 143 | 4% | 123 (86.0%) | 20 (14%) | 0.71 (0.45–1.19) | 0.174 | 0.278 |
| Bosch et al.— H37Rv[c] | | | | | | | |
| Essential | 677 | 18% | 644 (95.1%) | 33 (4.9%) | 2.29 (1.61–3.37) | 1.00e−05[g] | 9.03e−05[g] |
| Not essential | 3,062 | 82% | 2,740 (89.5%) | 322 (10.5%) | Ref | NA | NA |
| Bosch et al.—HN878[d] | | | | | | | |
| Essential | 654 | 18% | 621 (95.0%) | 33 (5.0%) | 2.19 (1.54–3.23) | 2.95e−05[g] | 2.66e−04[g] |
| Not essential | 3,084 | 83% | 2,762 (89.6%) | 322 (10.4%) | Ref | NA | NA |
| Gene set categories[h] | | | | | | | |
| *Mycobacterium* spp. orthologs | 1,006 | 26% | 985 (97.9%) | 21 (2.1%) | 6.64 (4.25–10.94) | 7.40e−27[g] | 1.18e−25[g] |
| In DosR regulon | 47 | 1% | 43 (91.5%) | 4 (8.5%) | 1.15 (0.42–4.5) | 1 | 1 |
| In enduring hypoxic response regulon | 208 | 5% | 191 (91.8%) | 17 (8.2%) | 1.22 (0.73–2.16) | 0.5463 | 0.583 |
| Genes whose disruption alters drug efficacy (mice) | 146 | 4% | 142 (97.3%) | 4 (2.7%) | 3.93 (1.49–14.72) | 0.002[g] | 0.004[g] |
| Genes with methylated promoters | 186 | 5% | 176 (94.6%) | 10 (5.4%) | 1.94 (1.02–4.15) | 0.0415[f] | 0.083 |
| Deletion mutants underrepresented in mouse model | 527 | 14% | 505 (95.8%) | 22 (4.2%) | 2.71 (1.74–4.43) | 6.97e−07[g] | 2.79e−06[g] |
| Toxins/antitoxins | 147 | 4% | 136 (92.5%) | 11 (7.5%) | 1.34 (0.72–2.78) | 0.4758 | 0.544 |
| PE/PPE genes | 149 | 4% | 113 (75.8%) | 36 (24.2%) | 0.31 (0.21–0.48) | 1.07e−07[g] | 8.59e−07[g] |
| Transcription factors | 199 | 5% | 188 (94.5%) | 11 (5.5%) | 1.88 (1.02–3.87) | 0.04773[f] | 0.085 |

[a]N = 3,766 (genes with missing essentiality data that belong to the core = 57 [80%]; belong to the accessory = 14 [20%]).
[b]N = 3,739 (genes with missing essentiality data that belong to the core = 82 [84%]; belong to the accessory = 16 [16%]).
[c]N = 3,837 (genes with missing essentiality data that belong to the core = 81 [83%]; belong to the accessory = 17 [17%]).
[d]N = 3,837 (genes with missing essentiality data that belong to the core = 82 [83%]; belong to the accessory = 17 [17%]).
[e]"Essential domain" included with the essential gene category due to low cell count.
[f]P-value < 0.05.
[g]P-value < 0.001.
[h]Genes from functional sets in core and accessory are not mutually exclusive (i.e., some genes may have duplicate copies in the core and accessory).
[i]NA, not applicable.

One of the main contributions of this project is a complete account of all *esx, pe, pe_pgrs*, and *ppe* genes, including their homologs and paralogs. Here, we report 12 *esx*, 26 *pe*, 45 *pe_pgrs*, and 45 *ppe* unique genes in the MTBC core genome, excluding duplicate copies (Table S1). We report an additional seven *esx*, six *pe*, eight *pe_pgrs*, and 16 *ppe* unique genes in the MTBC accessory genome (Table S1). To avoid reporting duplicates, these counts do not include genes that are merged by Panaroo (e.g., ppe12 ~~~ppe13). If one considers counts such cases, then the core counts increase to 17 *esx*, 28 *pe*, 50 *pe_pgrs*, and 52 *ppe* total genes, while the accessory counts increase to 14 *esx*, six *pe*, 11 *pe_pgrs*, and 29 total *ppe* genes. In our genomes, we observe the greatest variability in *esx* genes, in terms of presence and absence. However, this variability was mostly driven by L4-specific diversity (Table 1). *pe* genes were conserved across our genomes. We observed a total range (including duplicate gene copies) of 20–31 *esx* genes (median 23), 61–73 *ppe* genes (median 68), 30–33 *pe* genes (median 33),

and 54–63 *pe_pgrs* genes (median 68) within each genome (Table 1). For *esx* genes, L4 had marginally highest median but distinctly higher diversity than other lineages. While L2 had the highest median number of *ppe* genes among the first five lineages, L4 had a distinctly higher diversity in *ppe* gene content. The remaining lineages had a similar relatively lower *ppe* gene diversity (Table 1). In terms of the *pe_pgrs* gene content, L1 had the lowest median and L4 had the highest. In terms of diversity, perhaps unsurprisingly, the two most successful lineages L2 and L4 had a distinctly higher variability in terms of *pe_pgrs* gene counts (Table 1). Finally, the overall diversity of *esx*, *ppe*, and *pe_pgrs* genes was higher than the within-lineage diversity of each of these families, indicating the prevalence of lineage-specific genes in each of these families.

*Shannon's entropy* index is a measure of uncertainty (101, 102) used typically to discover information content (i.e., distinguish between noise and information). This index is shown in Data Set S4 with representative sequences in Data Set S5. The minimum index is zero indicating no change in the data. It elevates above zero when change is detected; however, it remains low if the change is random. It is elevated higher when the change follows a pattern (repeated observation). In this context, we used the measure to answer the question of which gene family's diversity (in terms of gene presence/absence) follows a conserved pattern (convergent evolution) and therefore carries the highest information content and hence is most impactful on the strains' phenotypic diversity. As can be seen (Data Set S4), Shannon uncertainty index of *ppe* genes is the highest at 4.377, followed distantly by the *esx* family at 2.494, *pe_pgrs* family at 2.073, and *pe* family at 0.588. The *pe* family was quite stable except for *pe27A* (*Rv3018A*, deleted in 33 isolates) and *pe25* (*Rv1787*, duplicated in 90 isolates). *pe_pgrs4* (*Rv0279c*, deleted in 54 isolates), *pe_pgrs30* (*Rv1651c*, present only in 17 isolates), *pe_pgrs49-50* (*Rv3345c-Rv3344c*, deleted together in 25 isolates), *pgrs54* (*Rv3508*, deleted in 23 isolates), *pgrs57* (*Rv3508*, deleted in 82 isolates, the remaining 27 had two copies of this gene), and minimally *pgrs28* (present in all isolates but four had a duplicate copy) had the highest contribution to *pgrs* Shannon index. Among *esx* genes, *esxJ, esxL, esxN, esxO* contributed the most to the convergent diversity (each change was observed in multiple lineages) in this family, with the latter three often duplicated together. Among *ppe* genes, *ppe34* (*Rv1917c*, greatest source of diversity through frequent duplication, often multiple times) and *ppe16* (*Rv1452c*, duplicated in 28 isolates) had the largest contribution. Even though *esx* genes show greater relative range of diversity, Shannon's entropy indicates that presence/absence of *ppe* genes carry more information and therefore are most impactful on the diversity of MTBC. None of the events reported in this section were lineage-specific and were observed convergently across multiple lineages (85).

## Putative novel genes are more prevalent in clinical isolates than in the H37Rv reference genome

Since putative novel genes were not found by Panaroo in H37Rv using the curated H37Rv GFF from NCBI as input, and thus clustered separately from other isolates, we hypothesized that identification of these putative novel genes was dependent on Prokka identifying the open reading frames. As such, we evaluated whether Prokka runs on the NCBI H37Rv-NC_000962.3 FASTA file would be able to identify the same putative novel genes present in our clinical isolates. Prokka identified 327 unnamed open reading frames in H37Rv, of which 77 open reading frames did not overlap with named gene coordinates in the curated NCBI GFF file. Alignment to our Panaroo-derived pangenome multi-FASTA file with BLASTN resulted in 79 matches. Genes with the highest query coverage, percent identity, and E-value were retained where queried sequences had aligned to multiple genes. In total, 74 open reading frames in H37Rv aligned to our pangenome's putative novel genes, two open reading frames aligned to named genes (*Rv2807, dsx*), and one open reading frame did not match any genes in the pangenome. Most of these unannotated gene regions in H37Rv aligned with high percent identity (median = 100%, range = 97.87%–100%) and sequence coverage (70/74 aligned at 100% coverage). Only four sequences matched at <100% coverage (median: 70.80%, range:

24.34-87.64%). These findings support our hypothesis that many genes identified by Panaroo and missing in H37Rv were due to using the NCBI curated GFF file as input to Panaroo, and not the Prokka-produced version. These findings spotlight the areas of the H37Rv file from NCBI that are yet unannotated, distinguishing them from truly novel genes in the MTBC pangenome (i.e., genes absent from the reference genome, 216/290 genes, 74%), and denote how relying on referenced-based annotations alone will likely miss some genetic content. It is important to also note that most isolates had at least 171 putative novel genes, which comprised at least 4.25% of their genomes. Seeing that H37Rv still had a much smaller number ($N$ = 74 matched) and percent (1.9% of total genome) of putative novel genes than the other isolates signifies that H37Rv lacks genetic content that is present in clinical isolates (Data Set S6).

## Characterization of putative novel genes show conserved regions of unknown function

Our principal component analysis highlighted how the presence or absence of putative novel genes may drive lineage clustering and were fairly conserved (Fig. 1; Table S2). We hypothesized that Panaroo-identified putative novel genes were likely gene fragments since they generally were short (median: 218 base pair [bp], range: 90–2,746 bp). Therefore, we aligned the Panaroo-derived representative sequences of these genes to the NCBI H37Rv genome to assess the proportion of putative novel genes that are gene duplicates, fragments, or unable to be determined. We assessed the best BLASTN hits with the highest percent identity and highest coverage (i.e., alignment length/query length) (Data Set S6). Of the 290 putative novel genes identified in the pangenome, 167 (58%) putative novel genes were able to roughly align to a named gene in H37Rv (164/167 [98%] were truly novel genes and entirely absent from the H37Rv reference genome). For 136 putative novel genes, the gene in H37Rv that aligned with the highest percent identity also had the highest query coverage where multiple BLAST hits were seen. Among these 136 putative novel genes that had concordant matches on identity and coverage, all exceeded 80% identity (median: 100%); however, the query coverage was often low (min: 1.48%, median: 23.18%, max: 102.13%, IQR: 11.20%–44.02%).

Twelve putative novel genes matched to a gene in H37Rv with high identity and coverage ≥80%, indicating that they are likely duplicates. While query coverage remained relatively high, the length of the gene in Panaroo exceeded the length of the aligned gene in H37Rv by ≥100 bp for five genes. The functions of most of the aligned genes remain unknown; however, some are hypothesized to encode fatty acid CoA ligase (*fadD18* [*Rv3513c*]) and a two-component system sensor kinase (*Rv2998A*). Two putative novel genes (group_266 and group_361) aligned with 100% identity and 92%–94% coverage to *fadD18*. Seeing that most of these putative novel genes are prevalent in our sample ($N$ = 6/12 conserved in the core genome; 9/12 present in over two-thirds of our isolates), it is important to further investigate whether these genes are being expressed and what role they may play in MTBC survival.

Of the 290 putative novel genes, 123 did not align with a gene by BLASTN. Of these, 76 genes were conserved in the core genome (18/76 [24%] were truly novel genes and not present in the H37Rv reference genome), constituting roughly 58% of all core putative novel genes. The genes that did not align did not differ greatly in sequence length (median: 186 bp vs. 255 bp in aligned genes). Gene ontology analysis of aligned genes revealed that general cellular processes were underrepresented (0.52-fold enrichment, FDR adjusted $P$-value = 0.023), and over-enriched for unclassified molecular function (1.57-fold enrichment, FDR adjusted $P$-value = 0.038), especially among aligned genes that are in the core pangenome (1.73-fold enrichment, FDR adjusted $P$-value = 0.018). These findings collectively suggest that while few putative novel genes may be duplicates, most are mutated fragments of genes of unknown function. As previous studies have shown that translation of mutated and truncated genes often retains some function (103), sometimes with clinically relevant changes (104), further

research investigation of these mutated fragments' function and role during infection is warranted, prioritizing those in the core genome.

Many putative novel genes were found in only one lineage (82/290, 28.3%). Only one putative novel gene was seen in all L3 isolates, one gene was in all L4 isolates (except for H37Rv), two genes were in all L5 isolates, and three genes were only seen in our L6 isolate (Data Set S6). The putative novel genes found only in L5 and L6 matched an annotated gene present in H37Rv with high percent identity (>99%) but low coverage (<25%), including a putative novel gene with high percent identity to a portion of IS6110 at a locus in the pangenome unique to all L5 isolates. More research with expanded data sets is needed to confirm the use of these regions as potential lineage markers.

## Gene duplications of drug-resistant and virulence genes are common and may aid drug resistance

To further explore mechanisms of gene gain, we assessed how many genes were duplicated across the pangenome. A total of 86 genes were duplicated in the pange-nome of all 110 isolates, adding 144 extra gene copies, constituting approximately 5% of the pangenome (Table 3). Genes with the highest average copy numbers included IS6110, IS1081, and *esxJ* (*Rv1038c*) (Table 4). However, some of the copies of *esxJ* were mis-annotations of *esxK* (*Rv1197*) since the two are nearly identical (six SNPs different) and the same length. While Panaroo corrects for most gene mis-annotation, Panaroo might not distinguish closely homologous genes well. The persistent challenge to correctly annotate these genes highlights the need for better annotation tools.

Most duplicated genes (56/86 genes) had a single gene copy (i.e., core copy) and less than 99% of isolates had duplicate copy/copies (i.e., accessory copy). Ten duplicated genes had all copies in the core, while 20 genes had all copies in the accessory (Fig. 2; Table 3). Core duplications comprised transposases (2/10), and sulfur metabolism (2/10), among other functions (amidase, ESX-1 secretion protein, magnesium transport protein). Duplicated genes in the accessory were not enriched for any particular function. Most duplicates were not tandem but duplicated elsewhere in the pangenome; however, six genes also had tandem duplicates in the pangenome neighborhood structure (IS1081, IS6110, *pe_pgrs57, esxJ*, and *cysA2* [*Rv0815c*]).

For lineage-specific pangenomes, duplicated genes were considered "core" if all isolates in the lineage had the duplication, and accessory if the duplication was in at least one but less than all isolates in the lineage. L4 had the fewest duplications where all gene copies were in the lineage-specific core genome, whereas L1 had the most genes with all copies in its core (Fig. S4; Table S4). The median number of duplicates differed between lineages (Kruskal Wallis, *P*-value = 0.0064), and were more frequent in L3 than in L4 (Fig. S5, *P* = 0.014). While some genes were consistently duplicated across all lineages, the number of duplicate copies and genes with the duplication varied across lineages. There were variable numbers of copies across lineages for IS element genes (IS1081 and IS6110) and *esxJ* (Fig. S5). L6 and 7 had fewer copies of IS6110 genes compared to other lineages, but not for IS1081. In general, modern lineages (L2, L3, and L4) had more IS element copies and more duplications among *pe* and *ppe* genes. However, ancient (L1 and L5) and intermediate lineages (L6 and L7) have more prevalent duplications of MmpL family genes than modern lineages (Fig. S5).

TABLE 3   Distribution and frequency of duplicate genes in core and accessory genomes

| Duplicated gene category | Number of genes |
| --- | --- |
| Total duplicated (% pangenome) | 86 unique/230 total (5%) |
| Total core (% pangenome) | 83 (36%) |
| Total accessory (% pangenome) | 147 (64%) |
| All copies in the core | 10 unique/20 total |
| All copies in the accessory | 20 unique/70 total |
| Copies in core and accessory | 56 unique/140 total |

**TABLE 4** Average copy number of commonly duplicated genes

| Commonly duplicated genes | Average copy no. |
|---|---|
| *ppe59*; *ppe57* | 1.83 |
| *pe1* | 1.98 |
| *Rv0036/Rv0515* | 2 |
| *mgtE* | 2 |
| *tuf* | 1.57 |
| *Rv0795* (IS6110 (1)) | 13.33 |
| *Rv0796* (IS6110 (2)) | 15.11 |
| *sseC1/2* | 2 |
| *cysA2/3* | 2 |
| *esxJ* | 3.28 |
| *Rv1042c/Rv1149* | 1.99 |
| *Rv1047* (IS1081) | 4.19 |
| *embR* | 1.87 |
| *Rv1313c* (IS1557) | 2 |
| *ppe34* | 1.61 |
| *Rv2082* | 2.13 |
| *ppe38* | 1.56 |
| *pe25* | 1.82 |
| *Rv2807* | 1.99 |
| *Rv3348* (IS1608) | 1.96 |
| *amiD* | 1.99 |
| *espI* | 1.99 |

A few of the duplicated genes in the pangenome are known to be associated with drug resistance, such as *mmpL5* (*Rv0676c*) and *embC-A* (*Rv3793-Rv3795*), affiliated with bedaquiline and ethambutol resistance, respectively. Duplications of *mmpL5* and *mmpS1* (*Rv0403c*) were observed in all L1 isolates and thus considered core for that lineage. Only two of these isolates had a fragmented copy of *mmpL5*. Upon further investigation, while the amino acid sequence of the *mmpL5* duplicate only had 32% identity to the copy in H37Rv, the E-value was very significant (8.34e−159), had a similar length (945 vs. 964 amino acids), and was predicted by AlphaFold (106, 107) to produce a MmpL5-like protein. Similarly, some genes in the *embC-A* operon had duplicate copies in isolates across all lineages. While duplication of *embR* (*Rv1267c*) was very prevalent (*N* = 96 isolates), this duplication also had a low percent identified by BLASTp (57%) but significant E-value (8e−156) and protein folding prediction of an EmbR protein. Only one isolate (1-0110) had a duplication of all genes in the *embC-A* operon. This isolate was found resistant to all tested antibiotics and thus labeled as XDR. None of these gene copies were found to have any intragenic variants; however, one copy of *embC* was shortened (1,550 bp, 47% of gene in H37Rv).

## Fragmentation-causing variants specifically can lead to pseudogenization and provide a biological advantage

To characterize gene loss, we evaluated the number of potentially pseudogenized genes caused by gene fragmentation. These fragments often occur through a mutation that produces an early stop codon that putatively disrupts translation. While Prokka can annotate open reading frames in prokaryotic genomes, it does not accurately label all gene fragments as arising from the same gene. Instead, it often separates fragments as distinct unnamed open reading frames and describes them in the note field as a "hypothetical protein." Panaroo's gene adjacency network corrects for these labeling errors by grouping fragments that arise within the same gene locus and reporting multiple Prokka locus tags at a given locus. We assessed the genes with multiple open reading frames at a given locus to delineate the distribution of fragmented and thus

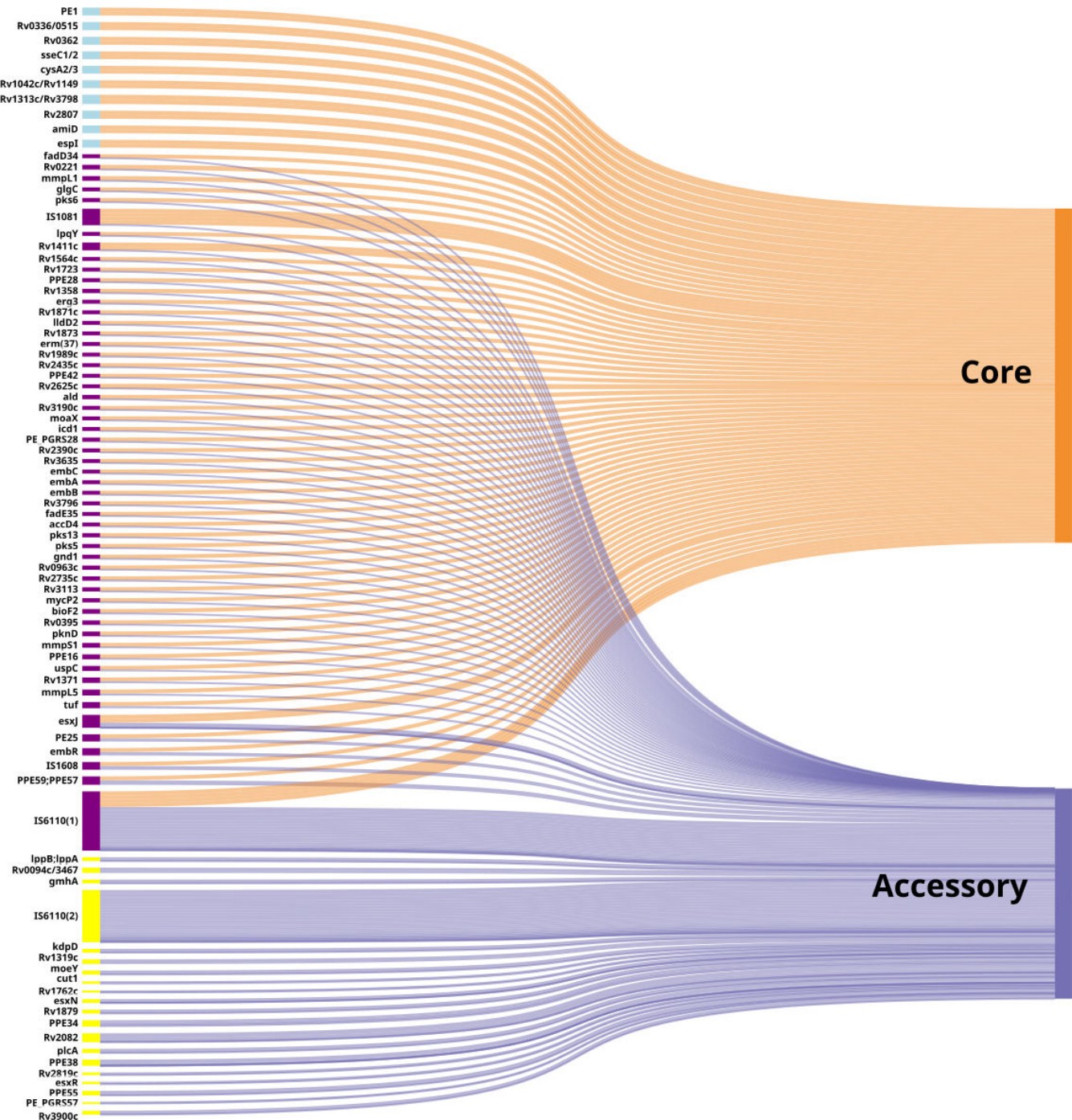

**FIG 2** Sankey diagram showing the distribution of genes with duplicate copies in the pangenome by the number of sequences in the core and/or accessory pangenome. The thickness of the line indicates a greater number of sequences. Sankey diagram made on SankeyMATIC (105).

potentially pseudogenized genes in the pangenome. Our pangenome analysis identified 667 genes (15.4% of pangenome) that had several Prokka-specific locus tag names in a single locus for at least one isolate. Of these 667 genes, 102 were merged annotations representing multi-gene loci, six were putative novel genes, and 559 named genes (554 unique genes); together, the 559 named genes and the 6 putative novel genes comprised 565 putative fragmented genes. Fragmented genes commonly occurred in isolation, with 294 out of 565 fragmented in only one isolate, and 495 genes fragmented in 10 or fewer isolates (Fig. S6). Except for H37Rv which did not have any fragmented

genes in the NCBI version, there were an average of 26 genes fragmented in each isolate (min: 8, median: 24, max: 65, std: 9.72). L1 isolates had a significantly higher median number of fragmented genes compared to L2 (L1: 36 genes, L2: 25 genes, *P*-value = 0.0001), and L4 (L4: 17.5 genes, *P*-value = 7.7e−13) (Fig. S7). While PacBio SMRT sequencing does have increased indel error rates, the error rate at the consensus level has been reported as low (108). Therefore, although we cannot definitively rule out some fragmentation as a result of sequencing error, it is reasonable to conclude that these fragments cannot be solely attributed to sequencing errors, but mostly reflect genes truly fragmented in these genomes.

We further investigated genes most frequently fragmented to assess if mutations causing fragmentation could provide biological advantages or simply reflect genes individually undergoing negative selection. No genes were fragmented in all isolates; however, a few genes were frequently fragmented, including *Rv0061c* (66 isolates), *Rv0134* (*ephF,* 56 isolates), and *Rv0797* (IS1547, 51 isolates). These top three most commonly fragmented genes were all fragmented by nonsense mutations, though *ephF* also had a mixture of nonsense variants and indels causing frameshifts. None of these genes had any duplications to recover function. *Rv0061c* had a nonsense mutation at nucleotide position 139 in 36 isolates (*Rv0061c*_c139t), which constituted all but one of our sample's L2 isolates. The nonsense mutation within *Rv0061c* on the complement strand occurs 402 base pairs upstream of the start of *Rv0062* (*celA1*) (*celA1*_g-402a) on the positive strand, a gene encoding for cellulase that may bear significance for biofilm maintenance and drug tolerance (109). To assess whether this mutation disrupted the promoter region of *celA1*, we utilized the motif-variant probe to identify if sigma factor binding motifs for *celA1* were disrupted by this variant (110). We found that the *celA1*_g-402a variant caused the creation of a SigI_10 sigma factor motif (G[AGT][AGCT] [GCA][TAG][CA], or GDNVDM in terms of IUPAC nucleotide ambiguity codes). Extra-cytoplasmic function (ECF) sigma factors like SigI are important for response to adverse environmental stressors (111, 112). The creation of a SigI binding motif preceding *celA1* may allow rapid regulation of biofilm, thus aiding its survival.

## Structural variants contribute to genetic diversity in conjunction with single-gene events

Structural variations (SVs) (insertions, deletions, transpositions, and variable-tract duplications) can be studied using gene-adjacency information in network analysis. The gene adjacency graph of the MTBC pangenome demonstrates resolved adjacency minus some regions (Fig. 3A). Panaroo found 1,935 SVs among the 110 isolates (Fig. 3D) distributed along the walk of the network with uniform genomic concentration. Short SVs (five or fewer genes) output by Panaroo were hierarchically clustered for global patterns in their distribution among isolates. This revealed no clear pattern between SV presence/absence and drug resistance for PZA, RIF, INH, or any fluoroquinolone or injectable drugs (Fig. 3E). SVs organized according to lineage, except for H37Rv, which clustered apart from L4 isolates. The discordant clustering of H37Rv with lineage is likely explained by SVs accumulated during its prolonged serial passaging and *in vitro* evolution. The median number of structural variants in an isolate was 676, with the max number (*N* = 702) contained by an L2 isolate.

Single genes contributing to genetic diversity through aggregate formations were extracted from the SVs, and a dictionary of contributions was created. Of the 4,325 genes identified by Panaroo, 1,457 were involved in an SV across the pangenome. These genes contributed to an average of two SVs, but 687 genes contributed to up to 44 SVs. Many SV contributors were duplicated, namely *Rv3799c* (*accD4*), *Rv1872c* (*lldD2*), *Rv2608* (*ppe42*), *Rv2352c* (*ppe38*), *Rv0035* (*fadD34*), *Rv1989c*, *Rv2780* (*ald*), *Rv2819c*, *Rv1873*, *Rv2625c*, *Rv1879*, *Rv1844c* (*gnd1*), *Rv0221*, *Rv3190c*, *esxJ*, *Rv2318* (*uspC*), *Rv1213* (*glgC*), *Rv0113* (*gmhA*), *Rv2390c*, *Rv0402c* (*mmpL1*), *Rv2351c* (*plcA*), *embR*, *Rv0795* (IS6110), *mmpS1*, *ppe34,* and *mmpL5*. Interestingly, the top 20 gene contributors to SVs were

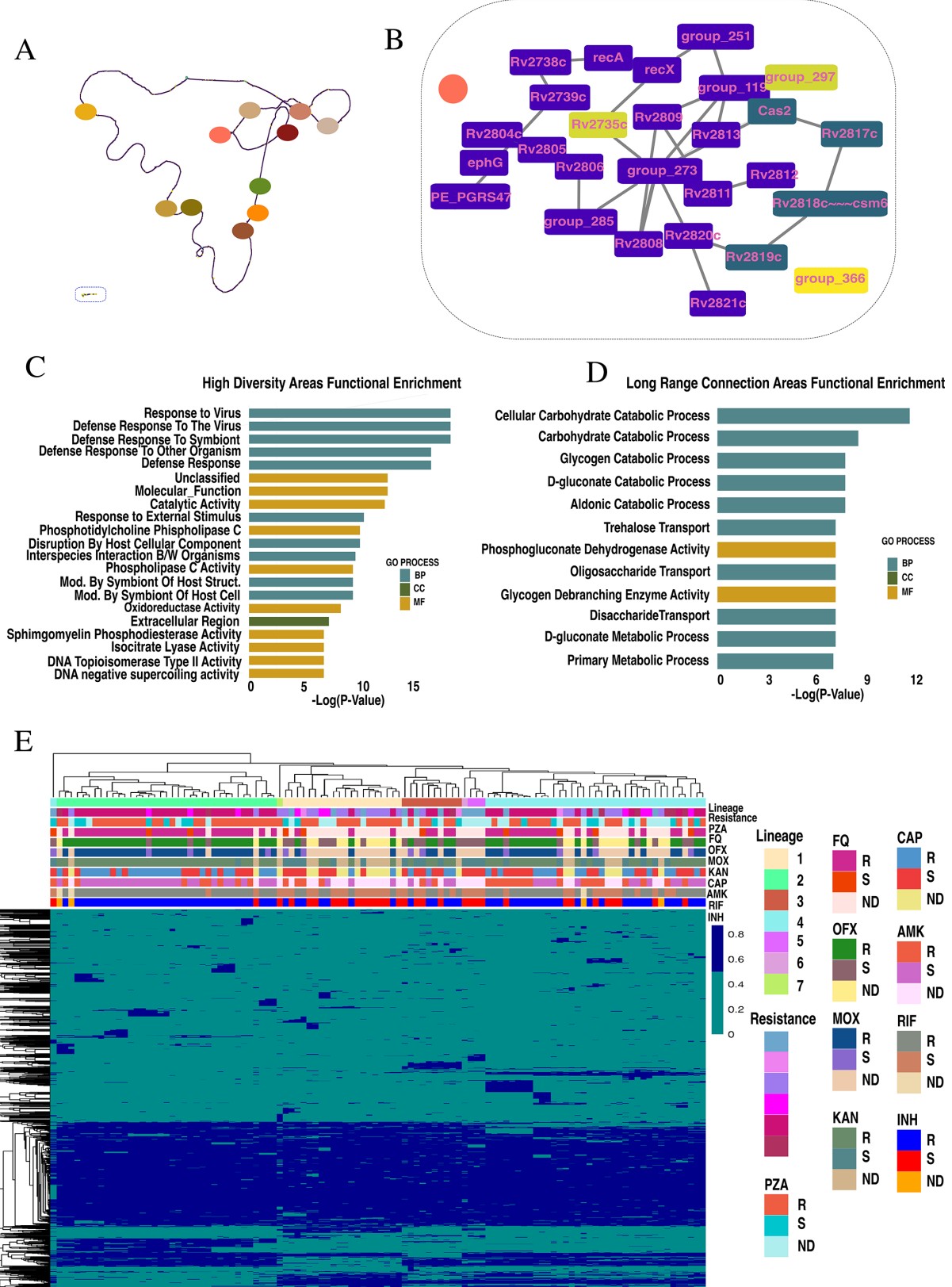

**FIG 3** (A) Topological overview of gene adjacency graph of *N* = 110 isolates in this pangenome analysis. (B) Gene-adjacency network of highly structurally variable region containing *cas2* gene-containing variants. (C) GO enrichment analysis of highly variable genomic regions found through orthogonal network (Continued on next page)

**Fig 3 (Continued)**

analysis. (D) GO enrichment analysis of long-range connections found through orthogonal network analysis. (E) Hierarchical clustering of Panaroo determined structural variants by Jaccard Index on presence-absence data. Annotation of drug resistance for pza, fq, ofx, mox, kan, cap, amk, rif, inh, lineage, and WHO classified drug resistance profile. (E) Counts of isolates according to antibiotic resistance (resistant, susceptible, or not determined).

putative novel genes (each contributing between 44 and 30 SVs), followed by *ppe39* (*Rv2351c,* 30 SVs) and an IS6110 element (29 SVs).

## Top contributors to structural variations associated with lineage and drug resistance include many putative novel genes and genes with merged annotations

An SV-genome-wide association study was performed to better capture the association between drug resistance phenotype and Panaroo-detected SVs. The model found SVs associated with drug resistance phenotype, but these results were nonsignificant after multiple hypothesis testing corrections (Table S4). The intersect of top hits before multiple hypothesis testing corrections, however, found SVs containing 96 putative novel genes ("group" genes) as well as 12 *ppe* genes, IS6110, 6 *esx* genes, 6 *pe_pgrs* genes, *Rv1266c* (*pknH),* and several *plc* genes (Table S5), supporting their importance not as singular genes, rather SVs in association with drug resistance. Eighty-seven genes were found in multiple SVs associated with drug resistance to any antibiotic, with the top prevalent genes including group_303, *Rv2348c,* group_217, the *esxN~~~esxO~~~esxL* merged annotation, the *lppA~~~vapB18~~~lppB* merged annotation and group_340.

The presence of the *esxN~~~esxO~~~esxL* merged annotation was interesting as it was not the separate genes called as significant in many drug-resistant SVs, but rather the multiple genes identified at that locus across genomes. This is a "merged annotation" with variable genetic composition among isolates. Figure S8 illustrates that these merged annotations (containing a tilde between gene names, e.g., geneA~~~geneB) were interchangeable at a given locus in the pangenome, different genes that were interchangeable with an added duplicate copy of one of the genes, multiple annotated genes present at the same locus, or an unnamed gene annotated as "*ab initio* prediction" by Prokka. L1 ($N = 18$), L2 ($N = 353$), L3 ($N = 75$), and L4 ($N = 335$, $N = 320$ when excluding H37Rv) contained SVs specific to the respective lineage (Fig. S9 and S10). L1 had 18 lineage-specific SVs, L2 had 353, and L3 included 75 variants (Tables S6 through S10). L4-specific variants (not including H37Rv) totaled 320 variants, adding H37Rv provided 15 additional variants (Tables S9 and 10).

## Systematic network analysis identifies larger structural variants than those reported by Panaroo

Complementary to Panaroo output SVs, a systematic network analysis was conducted to identify larger SVs. The gene-adjacency network (Fig. 3A) was not well interconnected as interpreted by the clustering coefficient ($0.1 \pm 0.2$), betweenness centrality ($0.0 \pm 0.01$), neighborhood connectivity ($2.7 \pm 1.9$), and topological coefficient ($0.5 \pm 0.1$) all being low (the value resting between 0 and 1; Fig. S11). Along the network, there are a total of 11 gene neighborhoods with high complexity in gene adjacency uncharacterizable by known network structures and contain an overrepresented number of nodes (2–3 std. deviations) (Fisher's exact test, *P*-value < 0.0001; Table S12; Fig. 3A and E). These regions house *pe*, *ppe*, and *pe_pgrs* genes, sites of IS transposon insertion, and consistently include many putative novel genes. GO term enrichment analysis of the high SV network regions identified enrichment in defense responses to virus or other organisms, interspecies interaction between organisms, modification by symbiont of host structure, molecular functions such as phosphatidylcholine phospholipase C activity, oxidoreductase activity, isocitrate lyase activity, and cellular component activity in the extracellular region (Fig. 3C).

In all, 191 genes contributing to Panaroo SVs were also found in this analysis. Many overlaps contributed to lower numbers of SVs, which is paradoxical to the high-order count and centrality of the genes/nodes in high SV regions. Five of the genes contributing to the most SVs were found in high structurally variable regions: group_122, group_290, *ppe39*, group_110, and group_134, contributing to 43, 37, 30, 32, and 42 variants, respectively.

When the backbone of the gene-adjacency network was resolved to an isolate dissimilar to the isolates included here (SEA11278), a total of 1,130 edges were cut, compared to the 4,057 that were conserved in the network. These long-range connections were through 69 nodes including IS element transposons, *pe/pe_pgrs/pe* genes, putative novel genes, and some ESX secretion system-associated genes. GO enrichment analysis rendered catabolic processes for cellular carbohydrate, glycogen, D-gluconate, aldonic acid, enzymatic activity for compounds such as phosphogluconate dehydrogenase and glycogen debranching, and some transport activity for trehalose, disaccharides, and oligosaccharides (Fig. 3D). The enrichment of these pathways was largely supported by the presence of genes *gnd1, Rv1564c* (*glgX*), *Rv1140, Rv2058c* (*rpmB2*), *Rv1358, Rv3327*, and Rv1527c (*pks5*).

The number of SVs found in these reference-quality genomes showcases the ability of these polished genomes to identify variations in gene adjacency across all contigs which have formed a resolved, contiguous genome. The genes that contribute to a low number of SVs are tributaries to low-level genetic diversity, complemented by those that contribute to a higher number of SVs.

## Pangenome analysis indicates MTBC has a slightly open pangenome

A comprehensive analysis of the pangenome was conducted to assess whether our MTBC isolates produce an open or closed pangenome. We conducted pangenome and core genome rarefaction simulations to fit the power law and exponential decay curves that comprise the pangenome curve (113). Figure 4 displays the change in gene clusters by step-wise random addition of genomes. The resulting Heaps alpha metric of less than 1 (0.983) indicates an open pangenome (114), which is supported elsewhere (29). This indicates that MTBC has an expanding genome, where increases in accessory genes are seen with the addition of more isolates. The alpha metric in our study is close to one, suggesting that the MTBC pangenome increases at a very slow rate.

## DISCUSSION

Our pangenome analysis showcases regions of the *M. tuberculosis* complex genome that exhibit the greatest genomic diversity across lineages. This pangenome was constructed from a large set of polished, reference-quality *M. tuberculosis* complex genomes, and was bolstered by two distinct forms of network analysis, supporting the robustness of identified structural variants implicated in drug resistance and virulence. Although most of the pangenome comprised highly conserved genes, genomic and network analyses showed genetic diversity through duplication events, gene fragmentation, creation of putative novel genes, and structural variation which could lead to changes in cellular phenotype (68, 115). Our findings also spotlight the need for more research into gene essentiality, as defined by H37Rv genes required for *in vitro* survival. Our finding that numerous *in vitro* non-essential genes are in the core genome, 38% of which provide functions known to be important for *in vivo* survival, suggests that the current definition of essentiality may underestimate and miss genes that have essential roles during infection or transmission within the host. More research is needed to clarify which genes are essential within these important *in vivo* contexts in humans, including genes that are not present in H37Rv.

The core genome falls between other reported core pangenome sizes, which range from 1,176 to 4,391 genes in sample populations comprised of 21 to >3,400 isolates. Other pangenome studies' foci were distinct from ours and included pangenomes among animal-host *Mycobacterium* strains (66), lineage-specific pangenomes (29), the

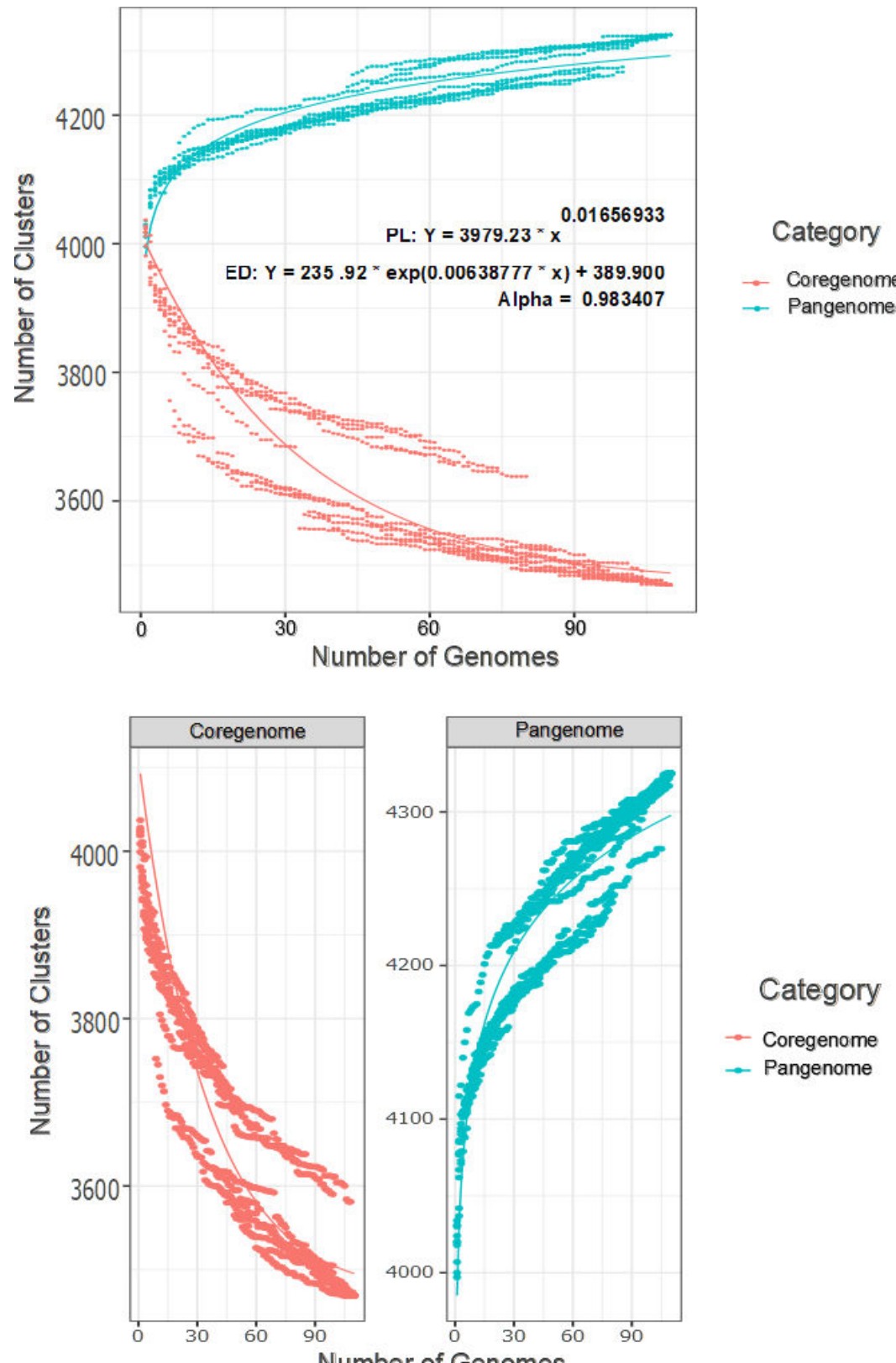

**FIG 4** Fitted power law (PL) and exponential decay (ED) curve for the pangenome rarefaction simulation. The attribute Heaps alpha < 1 indicates an open pangenome. Therefore, analysis of our sample of mostly drug-resistant isolates indicates *M. tuberculosis* complex pangenome is expanding at a very slow rate.

pangenome among modern and ancient lineages (55), and a pangenomes solely among *M. africanum* strains(Data Set S1) (30, 45, 57, 116). These pangenomes contribute a composite image of the state of genetic diversity among important MTBC populations and affirm many of the findings here including gene loss or duplication as a driver of virulence, host-tissue colonization, and the importance of metabolism in bacterial survival (particularly lipid metabolism). Our graph-based pangenome analysis of our in-house *de novo* assembled, long-read sequenced, reference-quality isolates captured a core genome size comparable to studies with much greater sample sizes, potentially due to improved annotation correction and capture of genes that previously were excluded due to shortcomings of short-read sequencing which results in annotation or alignment difficulties (54, 56); Table S1).

Our study found similar counts and composition, but higher diversity of *ppe*, *pe/pe_pgrs*, and *esx* family genes as have been described elsewhere (93). These genes are often excluded from studies due to their high sequence homology (35), frequent gene duplication and conversion, and large repetitive regions, which each complicate assembly and annotation (56, 117–120). Although *esx* genes carried the greatest relative diversity, the diversity of *ppe* genes carries more information. The greatest contributor to the diversity (among all genes) was *ppe34* which was frequently duplicated (often multiple times). *ppe34* is known to be highly polymorphic, and a hotspot for recombination IS6110 insertion hotspot (121). These duplications likely impact phenotype; *ppe34* is a cell wall protein that modulates host immunity: *ppe34*-induced $T_h2$ cellular immunity is associated with increased expression of cyclooxygenase-2 (*cox-2*) in dendritic cells (122). As such, *ppe34* duplication enhances the cell's ability to induce *cox-2* expression in dendritic cells. Since *esx* genes secrete PPE proteins (123), it appears that *ppe* genes' diversity may be used in targeted responses (e.g., *cox-2* expression modulation) to specific host/drug pressures while *esx* diversity is employed more for broad-scale environmental responses. As *pe/pe_pgrs/ppe* family genes play an important role in host immune interactions (93–95, 124), specifically through immune evasion (47, 97, 125) from antigenic variation caused by SVs, and are the target of several vaccine candidates (126–132), these findings could impact the research of pathogen therapeutic targets for vaccine development. Due to this genomic variation, research must focus on continuing to type the diversity of these genes before they can be considered for immune therapeutic purposes against infection.

Network analyses showcase other genes important for interaction with the host immune system and highlight areas of further research. The highly variable network regions which stemmed off the shared backbone (i.e., omnipresent and structurally conserved genes among our isolates) contained a GO enrichment for defense against phages/viruses, reported through overrepresentation of *Rv2816c* (*cas2*), *Rv2817c* (*cas1*), *Rv2818c* (*csm6*), *Rv2819c* (*csm5*), *Rv2820c* (*csm4*), and *Rv2821c* (*csm3*). *cas2, cas1*, and *csm6* are predicted to be in an operon, as are *csm5, csm4,* and *csm3*. *cas1* and *cas2* participate in a highly variable network region (breaking the operon structure) identified through systematic network analysis. *cas2* is reported in several structural variants implicated in fluoroquinolone (*cas1-cas2*-group_117) and moxifloxacin (*cas1-cas2*-group194) resistance in the Pyseer model executed. These genes center on a secretion system and induce IL-6, IL-1β, and TNF-α release in macrophages altering host immunity (133). These highly variable regions in the network also contain phospholipase C activity via the *plc* family of genes (corroborated through our orthogonal analyses), which induce apoptosis in macrophages. Any isolates with these genes (and sometimes their duplicates) could experience greater resistance to macrophage microbiocidal activity (134). More research needs to be done to elucidate the nature of the association between many SVs in gene order and drug resistance. Also, more investigation is needed to assess whether variations in gene order and copy number of *plc* genes may attribute a survival advantage and aid virulence through operon restructuring, or simply reflect genomic changes that occur upon culturing *in vitro* (135). These genes have been repeatedly

implicated in gene loss events and variation in clinical isolates (136, 137), substantiating the validity of our network analysis approach.

Higher-order structural variants similarly identified genes associated with host tissue colonization and immune system evasion in a few clinical isolates. The prominent trehalose transport pathway is differentially distinct among our isolates and contributes to caseous granuloma catabolism processes, macrophage lysosome-phagosome fusion, IL-12-mediated modulation of IFN-γ expression which centrally facilitates canonical granuloma formation, and the development of post-primary disease due to the high lipid content. The enrichment of lipid metabolism processes in the core genome through GO analysis highlights the centrality of functions that sustain the bacteria in-host within macrophages (48) and granuloma, the immune-cell environment that encases infection in the lung. This phenomenon as supported by these two orthogonal analyses has not been specifically named in previous pangenome analyses as an important delineating factor among isolates, despite these processes contributing to the high virulence of MTBC which can vary among isolates and even lineages. The long-range connections separated from the network backbone, signifying genes that are only in a few isolates and not structurally conserved, also show high enrichment in glycogen, glucan catabolism, and metabolism, which are important for bacterial cell wall maintenance and interaction with host cells during colonization of host tissues. Cellular carbohydrates form part of the pathogen's envelope and promote pathogenicity and virulence due to their protective effects from the host. This allows MTBC to remain in the host for extended periods of time and contributes to persistence (138). Since MTBC's ability to control the environment and immune system enables the pathogen to persist and infection advance, it is pivotal to evaluate whether genomes with an increased number of genes engaged in immunomodulatory processes, such as cell envelope or trehalose production, may produce more clinically severe disease.

Our pangenome analysis also shows gene duplication as a potential mechanism for drug resistance and tolerance. Our analyses revealed not only that an MmpL5-like protein is duplicated in several isolates across lineages but that this duplication was core in the lineage 1 pangenome. *mmpL5* has long been implicated in bedaquiline and clofazimine resistance through its ability to form a complex with membrane fusion protein mycobacterial membrane protein small 5 (MmpS5, encoded by *Rv0677c*) to transport drug compounds outside of the cell (139). Although overexpression of this operon through mutation in its repressor (*Rv0678*) has been shown to cause drug resistance, other mechanisms of overexpression such as duplication of genes and/or their paralogs have not been explored previously (140, 141). Importantly, given that the resistance gained by mutations in *Rv0678* is lost if there is a loss of function in *mmpL5* (142), it is also essential to assess whether the presence of an additional MmpL5-like protein may restore resistance to bedaquiline or cause drug tolerance. In addition, phenotypic consequences of changes in the expression of *mmpL1* or *mmpS1* (genes also important in transmembrane transportation) have not been investigated. Given that the MmpL gene family encodes for multidrug efflux pumps and undergoes partial if not whole gene duplication (143), it will be important to assess clinical and phenotypic relevance with increased expression of genes in the resistance-nodulation-cell-division superfamily (RND family) (144).

Similarly, analysis of gene duplications in one extensively drug-resistant isolate in our sample set revealed a duplicate copy of each gene in the *embC-A* operon which is implicated in ethambutol resistance. These genes encode for arabinosyl transferases which are important for arabinan biosynthesis needed to produce arabinogalactan in the cell wall (145). Although this was seen in only one isolate, 96 isolates had a duplication of an EmbR-like protein. The transcription factor *embR* is hypothesized to regulate the *embC-A* operon through its phosphorylation by PknH (146), as well as potentially regulate response to hypoxic stress (147). Duplications of these genes may provide a physical means of resistance or tolerance through increased cell well density, as noted elsewhere (148). Also, gene duplication may compensate for loss

of function mutations and confound molecular diagnostic tests. Overall, gene duplications may provide another method to confer drug resistance or tolerance, as increased gene dosage may enhance the function of drug efflux pumps and cell wall synthesis. Duplications and duplications of their paralogs may compromise molecular diagnostic testing, which will need to adapt not just for single nucleotide polymorphisms but also gene copy numbers or compensatory proteins.

While our pangenome analysis and others assert that MTBC may have an open pangenome, some studies have concluded the opposite. There are several contending facts that complicate calling the pangenome open or closed. MTBC strains are clonal species with minimal diversity compared to other bacteria, making a closed pangenome plausible. MTBC strains have no horizontal gene transfer, which differs from the canonical open pangenome seen in bacteria with horizontal gene transfer, such as *E. coli* (114). However, our analyses show that MTBC exhibits diversity through gene duplication events and structural changes in gene order which likely aid survival and pathogenic success (68). Also, roughly 5% of the clinical isolates' genomes were comprised of unnamed *ab initio* predicted genes (putative novel genes), giving more support to an open pangenome. Seeing that other pangenome studies with greater sample sizes (>1,000 isolates) reported slightly smaller core and larger accessory genomes than our study (54, 56)(Fig. S11), we conclude that the MTBC pangenome likely is open, despite having a slow rate of increase as indicated by our Heaps alpha metric.

Our analyses also highlighted important differences between our diverse set of clinical isolates and the H37Rv reference genome. While the proportion of putative novel genes remained consistent across clinical isolates, H37Rv contained a smaller number and proportion of these genes. These differences resulted in H37Rv clustering separate from all other isolates, including other lineage 4 isolates. Removing H37Rv from the L4 lineage-specific pangenome analysis increased the number of genes shared across all L4 isolates from 3,678 to 3,826 genes. This increase in L4 core genes suggests the genomic composition of H37Rv differed from other isolates within the L4 lineage but does not solely explain the difference in diversity seen in L4 compared to other lineages. The notable difference between H37Rv and the rest of L4 may be attributed to the fact that H37Rv has become a laboratory strain for the past several decades and has undergone a separate evolutionary path. This underscores the genomic differences in H37Rv from clinical isolates acknowledged by others (29, 40). One study of publicly available clinical isolates showed that H37Rv did not reflect the genomic structure observed in hypervariable regions, such as the *ppe38* region. A study of Beijing lineage isolates found 212 genes in their genomes that were not in H37Rv (149, 150). These studies and ours emphasize the caution of utilizing H37Rv for reference-based alignment of whole genomes, as a model strain for understanding TB infection in the host and spotlight the immense knowledge gap regarding the functional role of the putative novel genes in the MTBC genome.

A common theme among genes contributing to *Mtb* pangenomic diversity is frequent insertion by IS6110. The *ppe* genes with the highest information content (*ppe16* and *ppe34*), the genes identified by pangenome network analysis (the *Rv2816c-Rv2821c* and *plc* genes), and several resistance-associated structural variants all occur near, and suggest, IS6110 transposition. Importantly, not all genetic diversification that drives infection outcomes are captured by consensus sequence in cultured isolates; only those that are expelled and successfully transmitted to others will be captured. Isolates with a higher frequency of genomic compositional change will more frequently evolve subpopulations, some of which will contribute to worse infection outcomes. Our observations here are interesting in the context of Beijing isolates given their higher IS6110 copy number, their rapid global spread, and worse clinical outcomes for patients infected by them.

Our study contained a few limitations. Our study focused on *in silico* pangenome analyses and therefore requires further experimental confirmation of the effect of our findings on clinical outcomes and drug resistance. Because our data set is heavily

representative of isolates from lineages 1, 2, and 4 and drug-resistant phenotypes, the pangenome does not reflect the entire *M. tuberculosis* complex species. The fixed effects model regressing the Panaroo structural variants on antibiotic resistance profiles did not reach statistical significance, likely due to sample size and an imbalance of resistant/susceptible isolates (Table S13). In addition, Panaroo's gene reporting depends on the annotated inputs from Prokka to construct the pangenome. Using different annotation tools or parameters may produce different gene boundary calls, and thus may slightly vary in the number of genes identified, especially of the putative novel genes and gene fragments. Although Panaroo corrects for inaccurate gene labeling, it may struggle to identify genes with significant mutations and gene conversion events. Also, our estimation of an open pangenome does not consider the genes that are fragmented and thus potentially pseudogenized. While gene fragmentation adds to genetic diversity, not taking these into account could potentially overestimate pangenome growth. Lastly, given the high prevalence of frameshift mutations in the fragmented genes, we cannot rule out the possibility of sequencing error accounting for a small portion of our observed gene fragments. However, these genomes were sequenced with high depth (~100× on average) and used accurate P6C4 chemistry. PacBio continuous long-read sequencing is touted to maintain a low consensus error rate (108). Therefore, we can be confident that the observed gene fragmentation remains highly common.

## Conclusions

Our pangenome characterization of 109 *de novo* assembled genome of *Mtb* clinical isolates (and virulent type strain H37Rv) demonstrates conserved and variable regions among *M. tuberculosis* complex isolates. The variable regions can be attributed to manifold diversifying mechanisms, but IS6110 transposition was implicated especially frequently, particularly for diversifying genes with clear links to clinically pertinent phenotypic variation. This suggests that structural variation, and particularly IS6110 transposition, contributes to the high adaptability of this pathogen more so than currently appreciated. The pangenome is expanding, yet at a very slow rate, due to genetic duplications (including drug-resistant genes), fragmentations, and highly diverse structurally variable regions, which contribute to the balance of gene loss and gain. The slow rate of pangenome expansion can be explained by a genome that is mostly stable but highly dynamic and expanding in specific loci, such as *pe/ppe* genes. Clinically important areas of genetic variation included genes involved in host immune evasion (*Rv0071-73, Rv2817c, cas2*) and drug resistance (*mmpL5, embC-A*), which have significant implications for the development of vaccines and molecular diagnostics targeting these genes and their products.

## MATERIALS AND METHODS

### Pangenome analysis: preparation and global characterization

A pangenome was constructed using 109 phenotypically diverse (in terms of antibiotic resistance, Table S13, Set S7) clinical isolates of *M. tuberculosis* complex infection collected from the World Health Organization Supra-national References Laboratories in Stockholm, Sweden and Antwerp, Belgium, and the Global Consortium for Drug-resistant tuberculosis Diagnostics (GCDD) (PRJNA555636 and PRJEB8783), which have been previously described (81, 82, 88, 151). These sequencing data were collected between 2015 and 2019 on the Pacific Biosciences RSII instrument using P6-C4 chemistry and we previously assembled many of them in these aforementioned works. More details on genome assembly and quality control analyses are in the supplemental methods. Although four isolates have increased deletion rates compared to other isolates, these genomes were retained in the analysis since they met all other QC standards and did not differ greatly in our pangenome analysis (Fig. S12). Lineages were called using TBProfiler version 4.1.1 and database version 7fc199d (152). The GFF annotation files of these *de*

*novo* assembled genomes were produced by Prokka version 1.14.6 using the following input parameters: --genus Mycobacterium --kingdom bacteria --rfam --rnammer --gram pos –usegenus –coverage 0.95 (153). Using the --proteins flag, we also provided Prokka with H37Rv protein sequences based on a recently curated annotation of this reference genome (154). With the addition of the H37Rv reference strain GFF annotation file from the National Center of Biotechnology Information (NCBI) (H37Rv-NC_000962.3), these files were input into Panaroo version 1.3.0, a pangenome analysis software that utilizes a graph-based algorithm to construct the pangenome based on gene adjacency (78). Panaroo was run with default parameters to produce an output file of gene presence and absence by isolating FASTA sequences of representative genes in the pangenome, a Panaroo-generated pangenome graph, and the sequence alignment of the core pangenome.

## Gene essentiality

To evaluate the distribution of genes experimentally identified as essential through *in vitro* transposon mutagenesis studies (TnSeq) and a CRISPRi study (87), supplemental data from DeJesus et al.(80), Zhang et al. (86), and Bosch et al. were merged with the Panaroo output. In accordance with DeJesus et al., essentiality was defined as a multilevel categorical variable (80). Similarly, essentiality according to Zhang et al. was a multi-level categorical variable (86); essentiality was also evaluated as a dichotomous variable according to the Bosch et al. definition. Essentiality was also evaluated as a binary covariate according to the DeJesus et al. definition, with "essential" being recoded to include "essential gene," "essential domain," and "growth advantage," while "not essential" included "growth defect," "not essential," and "uncertain." The proportion of binary essential genes in the accessory pangenome (i.e., comprised of genes that were in less than 99% of isolates) was also evaluated as a continuous variable.

Enrichment of genes implicated in the literature as functionally important or producing clinically relevant phenotypes from *in vivo* mouse models were also evaluated. The data sets used here include genes that are known transcription factors (155), produce toxins and antitoxins, are in *ppe/pe* gene families, have MamA, MamB, or MamC DNA methylation motif sites in their promoters (88), have effects on drug tolerance or resistance in mouse model (156), whose deletion mutants are underrepresented in TB mouse model (156), are in the enduring hypoxic response stimulon (89), are in the DosR regulon (89), or are conserved across other *Mycobacterial* species, specifically *M. leprae, M. smegmatis, M. avium,* and *M. abscessus* (157). Genes used in this enrichment analysis were not exclusively in either the core or accessory genomes. Instead, a core gene (i.e., in 99% of isolates) with an accessory duplicate could be represented both in the core and accessory genomes.

## Shannon's entropy

We used Shannon's entropy to identify which gene family's diversity (as defined by gene presence/absence) carries the highest information content and therefore has the potential for the most impact on the strains' phenotypic diversity. Shannon's entropy is calculated as follows:

$$H = -\sum_{i=1}^{S} P_i \ln(P_i)$$

where $S$ is the total number of genes in a family, $i$ is the gene index, and $P_i$ is the ratio of isolates whose genome harbors gene $i$ over a total number of isolates (101, 102). Gene duplications and merged annotations were included in this analysis.

## Principal component analysis and cluster analysis

We conducted principal component analysis (PCA) and unsupervised hierarchical cluster analysis to compare genomic profiles constructed from gene presence/absence data between isolates. Binary presence and absence data of genes across the 110 queried genomes was input to construct both PCA biplots and heatmaps, demonstrating the patterns of gene presence in the core and accessory genomes across lineages. PCA loading scores for individual genes and their respective Pearson correlation $r^2$ values were evaluated to assess which genes were most highly correlated to the principal components. All PCAs and figures were conducted using the PCAtools package in R (158).

## GO enrichment analysis

Functional gene enrichment of the gene clusters found in the core and accessory pangenomes was conducted using gene ontology (GO) analysis via PANTHER (159). Over-representation of biological process, molecular function, and cellular component GO terms present in the pangenome core and accessory genomes was assessed for significance using a Fisher's exact test at $P < 0.01$ and corrected for multiple hypothesis testing using a false discovery rate against the background of the $M. tuberculosis$ curated gene set collection hosted on the PANTHER database. These enrichments were also assessed by lineage to find potential functional distinctions of conserved function among isolates according to this factor.

## Putative novel genes

Prokka v. 1.14.6 was run on the H37Rv FASTA file from NCBI (H37Rv-NC_000962.3) with the same input parameters as the clinical isolates to identify which Panaroo-identified unique genes were, in fact, present in H37Rv as unnamed open reading frames. The open reading frames that were not assigned a gene name in the Prokka-produced GFF of H37Rv were extracted. The start and stop locations of these open reading frames were compared with the H37Rv-NCBI curated GFF file to filter out open reading frames that overlapped in nucleotide position with other annotated genes. The nucleotide sequences from the remaining unnamed open reading frames were aligned to the Panaroo-derived reference multi-FASTA file using BLASTN (160) to identify putative novel genes in our pangenome that are missed from the NCBI version of the H37Rv annotation.

We then sought to roughly characterize from which genes the putative novel genes may be derivatives as gene duplicates or gene fragments. Thus, Panaroo-derived sequences of these genes were all aligned to the NCBI H37Rv genome using BLASTN with default parameters. The best hits according to percent identity and query coverage were reported.

## Gene fragmentations and power law curve

Potential fragments were identified by Panaroo as having multiple Prokka-generated IDs in a single locus in the binary presence/absence output. Panaroo routinely combines fragmented genes as a result of a fragmented assembly (78). However, since our genomes were constructed with long read sequencing and a robust assembly pipeline, we conclude that these Prokka-identified open reading frames disrupting normal gene coordinates are likely to be genes fragmented by an early stop codon. The isolates and their genes with potential fragments were indexed, and their gene positions and lengths were compared to the corresponding gene length in H37Rv to identify which were tandem duplications versus fragments whose summed coverage equaled that of the copy in H37Rv. Since most putative novel genes were not in H37Rv, the lengths of these putative novel genes with putative fragments were compared to their average sequence length in the pangenome. Genes with multiple different genes annotated at a single locus (i.e., merged annotations, Fig. S8) were also excluded from this analysis.

A pangenome and core genome rarefaction simulation was conducted to estimate the number of genes in the core or pangenome given 100 random permutations of increasingly larger sample sizes of genomes (113). The pangenome rarefaction simulation was then used to construct the power law curve, while the core genome rarefaction simulation was used to fit an exponential decay curve. Collectively, these two curves are graphed to create the pangenome curves, which reflect the tradeoff between core and accessory gene cluster gain as the pangenome size expands with the stepwise addition of more isolates (113). The Heaps alpha value was calculated according to the Heaps law model described elsewhere (114) to inform if the pangenome is open (i.e., increases in genomic content with the addition of isolates) or closed (i.e., the number of genes do not continue to increase with more isolates, suggesting a limit to the bacterial genomic content). These analyses and graphs were constructed in R with the Pagoo package using default parameters (113).

## Network analysis

A fixed-effects linear model with a binomial family and a population structure correction using mash-calculated distances was conducted for structural variants output by Panaroo. The models were executed against binary drug resistance profiles (Resistant/Susceptible/Not Determined) for the 110 genomes in this data set using Pyseer v.1.3.10 (161). The phenotypes tested included the following: pyrazinamide, fluoroquinolone, ofloxacin, moxifloxacin, kanamycin, capreomycin, amikacin, isoniazid, and rifampicin resistance. Beta coefficients of all models were exponentiated to calculate the odds ratios, and confidence intervals based on beta coefficient standard errors were also calculated. A correction for multiple hypothesis testing was determined by assessing the FDR using p.adjust() in base R v.4.2.2 (162).

The adjacency-derived network graph's layout was simplified with the gene-adjacency information for isolate SEA11278, a monoresistant lineage 7 isolate (biosample SAMN12325265 bioproject PRJNA555636) that is most dissimilar (demonstrated through hierarchical clustering analysis; Fig. 1) to the other isolates present in this pangenome, then circularized for downstream network analysis. Using Cytoscape 3.9.1 (163), a standard network analysis approach was used to query the network for potential drivers of SVs by extracting areas of the genome where the gene neighborhood contained more than two nodes with more than two degrees. This threshold served as the distinguishing factor for local gene neighborhoods of high adjacency variation. Centrality measures for all nodes in the pangenome graph including global network measures were calculated using Cytoscape NetworkAnalyzer (164), which reported on network size (defined as the presence of nodes across all genomes included in this networks analysis), degree, average shortest path length, clustering coefficient, betweenness centrality, neighborhood connectivity, radiality, closeness centrality, eccentricity, and topological coefficient. Long-range connections (i.e., edges between the main pangenome network graph and unsupported nodes removed due to their trans-network connections) were characterized by size and functional enrichment using GO analysis through PANTHER. Networks were visualized and analyzed in Cytoscape, with all other data cleaning and analyses done in R4.1.0 (162), Rstudio 1.4.2.1, and Python 3.9.7 (165).

## Statistical analyses

Comparisons of genome size, lineage-specific pangenomes, and percent of essential genes across lineages were evaluated using one-way Welch's ANOVA and Games-Howell test for multiple pairwise comparisons due to unequal variances across the lineages. Kruskal-Wallis and Dunn's tests were used to evaluate statistical differences in the number of fragmented and duplicated genes across lineages, as data for these variables necessitated non-parametric tests. Since the H37Rv-NCBI GFF lacked the annotation of any putative novel genes and fragmented genes, it was identified as an outlier and removed from these bivariate statistical analyses. Lineages 6 and 7 were also removed

from these statistical tests due to low sample size. Lineage 5 isolates were also removed from Kruskal-Wallis tests due to low sample size.

As essentiality studies were defined in the context of genes in H37Rv, bivariate analyses were restricted to named genes within H37Rv, and thus we removed 488 putative novel genes and merged annotations (i.e., loci in the pangenome where different isolates had different and at times several genes called, Fig. S8). Also, we restricted the analysis to genes with no missing essentiality data (DeJesus et al.: 3,766/3,837 [98%], Zhang et al.: 3,739/3,837 [97%], Bosch et al.: 3,837/3,935 [98%]) or functional enrichment data (Table 2). Bivariate relationships for essentiality were evaluated using Fisher's exact test to assess enrichment in the core versus accessory genomes. *P*-values were adjusted for FDR due to the large number of multiple comparison tests. Simple logistic regression models provided quantified estimates of the size of enrichment, with model fit estimates used to determine the appropriateness of the models.

## ACKNOWLEDGMENTS

The authors acknowledge Derek Conkle-Gutierrez and Maryam Ahmadi J. for their informative discussions which contributed to our study. The authors also acknowledge Sarah Ramirez-Busby for her contribution to the quality assessment of this study's assembled genomes.

This work was funded through a grant (R01AI105185 and R01AI163202) by the National Institute for Allergy and Infectious Diseases (NIAID).

Conceptualization, Data Curation, Methodology, Investigation, Formal Analysis, Visualization, Writing—Original Draft, Writing—Review & Editing: M.E.E., A.M.S., and F.V. Supervision, Funding acquisition: F.V. QC of genomes, Writing: A.E.

## AUTHOR AFFILIATIONS

[1]Laboratory for Pathogenesis of Clinical Drug Resistance and Persistence, San Diego State University, San Diego, California, USA

[2]San Diego State University/University of California, San Diego | Joint Doctoral Program in Public Health (Global Health), San Diego, California, USA

[3]Department of Electrical and Computer Engineering, San Diego State University, San Diego, California, USA

[4]Department of Electrical and Computer Engineering, University of California San Diego, San Diego, California, USA

## AUTHOR ORCIDs

Monica E. Espinoza http://orcid.org/0000-0003-3577-7007
Ashley M. Swing http://orcid.org/0000-0002-5721-7150
Afif Elghraoui http://orcid.org/0000-0002-6489-9444
Samuel J. Modlin http://orcid.org/0000-0002-7674-305X
Faramarz Valafar http://orcid.org/0000-0002-3648-9384

## FUNDING

| Funder | Grant(s) | Author(s) |
| --- | --- | --- |
| HHS \| NIH \| National Institute of Allergy and Infectious Diseases (NIAID) | R01AI105185 | Monica E. Espinoza |
| | | Ashley M. Swing |
| | | Afif Elghraoui |
| | | Samuel J. Modlin |
| | | Faramarz Valafar |
| HHS \| NIH \| National Institute of Allergy and Infectious Diseases (NIAID) | R01AI163202 | Monica E. Espinoza |
| | | Ashley M. Swing |

| Funder | Grant(s) | Author(s) |
| --- | --- | --- |
| | | Afif Elghraoui |
| | | Samuel J. Modlin |
| | | Faramarz Valafar |

## DATA AVAILABILITY

Complete genomes for all 109 *M. tuberculosis* complex clinical strains are deposited under BioProject accessions PRJNA555636 and PRJEB8783. Custom codes for pangenome analyses are located at https://gitlab.com/LPCDRP/pangenome.

## ADDITIONAL FILES

The following material is available online.

### Supplemental Material

**Data Set S1 (mSystems00499-24-s0001.xls).** Published Mtb pangenomes.

**Data Set S2 (mSystems00499-24-s0002.xlsx).** List of all genes present in pangenomes.

**Data Set S3 (mSystems00499-24-s0003.csv).** List of all genes present in lineage 4.

**Data Set S4 (mSystems00499-24-s0004.xlsx).** Shannon's entropy scores of PPE, ESX, PGRS, and PE genes.

**Data Set S5 (mSystems00499-24-s0005.xlsx).** Unique, non-merged genes by gene family with their representative sequences.

**Data Set S6 (mSystems00499-24-s0006.xlsx).** Unnamed *ab initio* predicted genes identified by Panaroo.

**Data Set S7 (mSystems00499-24-s0007.xlsx).** Study isolate pDST, lineage, and NCBI Biosample accessions.

**Supplemental material (mSystems00499-24-s0008.docx).** Supplemental figures and tables.

### Open Peer Review

**PEER REVIEW HISTORY (review-history.pdf).** An accounting of the reviewer comments and feedback.

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
