## [Reviewer comments · mSystems]

Interred mechanisms of resistance and host immune evasion revealed through network-connectivity analysis of *M. tuberculosis* Complex graph pangenome

Monica Espinoza, Ashley Swing, Afif Elghraoui, Samuel Modlin, and Faramarz Valafar

Corresponding Author(s): Faramarz Valafar, San Diego State University

Review Timeline:

Submission Date:	April 5, 2024
Editorial Decision:	May 22, 2024
Revision Received:	July 13, 2024
Editorial Decision:	September 9, 2024
Revision Received:	December 4, 2024
Accepted:	December 16, 2024

Editor: Zackery Bulman

Reviewer(s): The reviewers have opted to remain anonymous.

Transaction Report:

DOI: <https://doi.org/10.1128/msystems.00499-24>

Re: mSystems00499-24 (Interred mechanisms of resistance, host immune evasion, and an incomplete perspective on essentiality revealed through network-connectivity analysis of *M. tuberculosis* graph pangenome)

Dear Dr. Faramarz Valafar:

Editor Comments: Thank you for submitting your work to mSystems. Please find below and attached the reviewer comments, which I agree with. I would like to provide you with the opportunity to revise the manuscript in accordance with these comments. Please provide point-by-point responses to the reviewer comments and make a notation of the specific line numbers where the corresponding modifications were made to the manuscript.

Revision Guidelines

Sincerely,
Zackery Bulman
Editor
mSystems

Reviewer #1 (Comments for the Author):

Espinoza et al. studied the pangenome of 110 de novo assembled clinical Mtb genomes. I have the following comments for the

authors' consideration

1. The authors should provide detailed genome sequencing and assembly methods. It appears that all genomes were sequenced using PacBio long reads. Did the authors implement a quality control step for genome sequencing and assembly? The accuracy of genomic analysis results can be significantly affected by low sequencing quality.
2. The gene essentiality analysis should be based on TnSeq or CRISPR-Seq experiments. Different strains from various lineages may have distinct sets of essential genes. It is not appropriate to infer gene essentiality based solely on the analysis of a single strain like H37Rv.
3. A major limitation of the study is its primary focus on sequence analysis without experimental confirmation of antimicrobial resistance.

Reviewer #2 (Comments for the Author):

The manuscript describes in silico evaluation of the pangenome of 109 long-read sequencing *M. tuberculosis* genomes. Several duplications involving antibiotic resistant, or in genes that might benefit the bacteria were found across different lineages of *M. tuberculosis*. Dense work was developed in order to try to understand deletions/SV/duplications, and how these events can affect the pathogen. Understanding how the MTBC complex can evade and cause disease, and understanding how the genomes evolve within hosts are crucial information, however I have few concerns with the manuscript. Please evaluate the attach file containing the review.

Comments to the authors

The manuscript describes in silico evaluation of the pangenome of 109 long-read sequencing *M. tuberculosis* genomes. Several duplications involving antibiotic resistant, or in genes that might benefit the bacteria were found across different lineages of *M. tuberculosis*. Dense work was developed in order to try to understand deletions/SV/duplications, and how these events can affect the pathogen. Understanding how the MTBC complex can evade and cause disease, and understanding how the genomes evolve within hosts are crucial information, however I have few concerns with the manuscript.

Main points -

- 1- The literature review was not complete, and in some sentences the authors tend to describe that there are not other studies. However, there are many other pangenome studies, even with more 100 genomes.
- 2- Lineage categorization was not well defined. The authors mention the occurrence of *Mtb* lineages 5 and 6, but they are caused by *Mycobacterium africanum*. There is no indication that other MTBC members were included in the manuscript. Also, how the lineages were categorized, by RDs?
- 3- Were any not *Mtb* genomes detected? Many times, some *Mtb* datasets include *M. bovis* by mistake, for example. That's why I recommend RD check – described by Warren et al., 2006.
- 4- Additionally, it's important to check for heterogeneous mutations - in case of contamination of the sample or even infection with 2 different *Mtb* strains - GATK, varscan, and others offer the option to evaluate heterogeneous calling.
- 5- The authors assume that the dataset used has high quality because the genomes are long read, but long read errors were not considered. And some sentences induce the reader to believe that all the other thousands of genomes available in NCBI have poor quality. Additionally, the authors are not considering the combination of short+long sequencing to prevent base calling, or SV/deletion errors.
- 6- Unique genes – long read sequencing can generate wrong calling, with miscalling small gene. The authors checked with blastn in only the dataset created in the study, or within NCBI dataset? That was not clear.
- 7- The bio projects used in the study mention 92 and 70 isolates, please clarify with a new table each isolate, adding specific information of each - SRA ID, lineage, atb resistance, and other pertinent information. Also, why these datasets were selected, against other studies that also used long read? Please clarify

Some details that can be better evaluated:

Line 19 – first time mentioning *Mycobacterium tuberculosis*; don't abbreviate

Line 20 - that is not the first study to use many complete genomes to evaluate pan genome, the authors are indicating that other studies were made with bad quality genomes? Please provide better data for that.

Line 39 - there are other old pathogens involved with human evolution, I recommend the authors to rephrase this sentence

Line 45 – *de novo* should be italicized

76 - Would be Mtb lineages instead of complex?

80 - that is not correct, please review literature

119 - there are thousands of mtb genomes, why the specific dataset used in the study would be better? Why not consider then, all NCBI datasets? The authors should compare with other data/genomes to suggest that only these genomes have the best quality

Line 131 - please, use comma for numbers (4,325) - nomenclature as line 479

141 - Careful with unique genes - specially older long read sequencing the creating of small gene fragments was a problem, leading to the use of illumina that would work better with high gc content mycobacteria. Please check those genes if they are really unique or just miss annotation

Lines 141-144 - that would be better fit in discussion

149 - when lineages were classified? and how?

154-158 – discussion.

Figure 1 - here the authors include *M. africanum* (lineage 5 and 6), these two species are not *M. tuberculosis*, but are in MTBC complex. The methodology only includes *M. tuberculosis* in the sampling - please clarify if different species from MTBC were included.

Line 161 – figure 1 legend - *n*= should be italicized, also, keep consistent, some lines are presented with “N”, some with “n”.

Line 174 – wouldn't be better to start results showing all the dataset, showing the lineages, antibiotic resistance and other features? Maybe at ~ line 145

Line 209 - first time introducing the species, please use full scientific name - *Mycobacterium leprae*, ...

Line 215 – check gene vs protein nomenclature.

Line 232-233 – reference to support the statement.

Line 234 – and other lines, please, include Rv gene ID among the studied genes

Line 236 – table S1 - please, make the sequence available.

Line 275 - Would be better to combine this topic when fig 1 is first introduced

Line 276 – wouldn't be among lineages?

Line 327 - what about other mtb genomes available in NCBI, were the genes ever described?

Line 358 - against what dataset? the one generated herein, or against all NCBI.

Line 420-421 - is there any data to support the drug resistant, or reference?

453 - reference to support the survival advantage.

Line 575 - the authors included *M. africanum*

Line 576-577 - Please explain why the picked bio projects would have better quality, compared to all the other dataset available in SRA;

It's known that the use of illumination + long read can give better genome, I recommend the authors to include those genomes to evaluate how is the quality of only long reads vs combine.

Line 583 – in vivo – italicize

Lines 588-590 - there are many other papers that the authors should consider to compare and to mention

- <https://journals.plos.org/plosone/article?id=10.1371/journal.pone.0175330>

- <https://www.frontiersin.org/journals/pharmacology/articles/10.3389/fphar.2018.01492/full>

- <https://www.mdpi.com/2079-6382/10/5/500>

- <https://www.mdpi.com/2076-2607/11/6/1495>

- <https://pubmed.ncbi.nlm.nih.gov/29259589/>

- <https://www.ncbi.nlm.nih.gov/pmc/articles/PMC9676053/>

Line 861 – explain what isolate is that - SEA11278; add table with all SRA ids, and different info.

Subject: mSystems00499-24 Decision Letter

Prompts in black text, Responses in blue

Reviewer #1 (Comments for the Author):

Espinoza et al. studied the pangenome of 110 de novo assembled clinical Mtb genomes. I have the following comments for the authors' consideration

1. The authors should provide detailed genome sequencing and assembly methods. It appears that all genomes were sequenced using PacBio long reads. Did the authors implement a quality control step for genome sequencing and assembly? The accuracy of genomic analysis results can be significantly affected by low sequencing quality.

We have clarified that the sequencing data was produced and described in our previous publications and added a brief description of the sequencing. We have already described the assembly and assembly quality control in the Supplemental Methods (in the Supplemental Figures_Tables.docx) and referred to it in section 5.1, lines 774-779.

2. The gene essentiality analysis should be based on TnSeq or CRISPR-Seq experiments. Different strains from various lineages may have distinct sets of essential genes. It is not appropriate to infer gene essentiality based solely on the analysis of a single strain like H37Rv.

We agree that the gene essentiality analysis should be based on TnSeq or CRISPR experiments. While inclusion of a more diverse set of isolates to assess true gene essentiality is ideal, we are limited by the data that is available from previous studies. Our review of the literature found 22 studies conducting TnSeq essentiality experiments all using H37Rv as the sole *M. tuberculosis* strain, with most referencing or comparing to the DeJesus et al. 2017 study, which we included in our analyses (PMIDs: 27407107, 25636614, 12657046, 35004921, 31239393, 21980284, 23028335, 24103077, 24315099, 23028335, 26438867, 26067605, 27936238, 28893793, 28504669, 30318462, 31363023, 14569030, 15928073, 36945430, 35112666, 16868085). This is likely due to the technically cumbersome and expensive nature of TnSeq experiments. The overwhelming use of H37Rv for TnSeq and CRISPR-interference (CRISPRi) experiments was also substantiated by a recent preprint (doi: <https://doi.org/10.1101/2021.03.05.434127>). Only three studies include clinical strains: hypervirulent strain HN878 (PMID: 34297925), *M. tuberculosis* GC1237 (PMID: 25956932), and one study with eight phylogenetically diverse clinical strains (PMID: 29505613). The study that assessed the eight clinical isolates evaluated genetic requirement for growth, as defined as differences in transposon mutant abundance that reflect differences in the fitness cost. They identified a small (~10-50 genes per strain) subset of genes with differential genetic requirements by comparing to H37Rv, which became the focus of their study. Therefore, the available data from this study is not in a comprehensive, genome-wide qualitative classification needed for our analyses. Similarly, the study using *M. tuberculosis* GC1237 (PMID: 25956932) focused on genes whose TnSeq mutants had fitness costs during survival in mouse dendritic cells over the course of 7 days, and themselves use the Zhang et al. 2012 study (which we also included in our analyses) to define essentiality.

Thus, we previously chose the two TnSeq experiments as the datasets for essentiality analyses (PMID: 28096490 & 23028335) due to their comprehensiveness, prominence in previous essentiality studies, and availability of compatible data. We have added wording to the methods to make it clearer that these essentiality calls were derived from TnSeq experiments. We have also added a third essentiality dataset (PMID: 34297925) from the aforementioned study using the hypervirulent HN878 strain through CRISPRi. We similarly evaluated enrichment of core genes among genes called essential in the CRISPRi dataset, both for H37Rv and for the HN878 isolate. This allowed us to evaluate whether conservation in the core genome matched essentially called by TnSeq or CRISPRi, and whether the core genome was more enriched in H37Rv-essential genes, or HN878-essential genes. This analysis has been added to our results and associated tables.

We do not contest the reviewer's assertion that essentiality in H37Rv does not necessarily imply essentiality in clinical isolates, and agree with the reviewer on this point. However, it is important to note that essentiality is largely conserved between clinical isolates—98% matching essentiality calls between H37Rv and the Lineage 2 clinical isolate HN878 (PMID: 34297925). Besides, our essentiality enrichment analysis was used to demonstrate that essential genes were more conserved than non-essential ones, and to generate a degree of enrichment to contextualize enrichment found in the other gene sets. We are not attempting to claim that any given gene is invariably essential in clinical isolates. Rather, we are describing the relative frequency of H37Rv-essential genes being invariably conserved (i.e., in the core genome) among clinical isolates.

3. A major limitation of the study is its primary focus on sequence analysis without experimental confirmation of antimicrobial resistance.

We thank the reviewer and completely agree with this limitation. Due to our limited access to a BSL3 laboratory space required for such experiments, we restricted our analyses to *in silico* evaluation with the goal of generating data-driven hypotheses that should be explored through subsequent studies. To make this limitation clearer, we have added this sentence to our discussion beginning in line 736: "Our study focused on *in silico* pangenome analyses, and therefore requires further experimental confirmation of the effect of our findings on clinical outcomes and drug resistance."

Reviewer #2 (Comments for the Author):

The manuscript describes *in silico* evaluation of the pangenome of 109 long-read sequencing *M. tuberculosis* genomes. Several duplications involving antibiotic resistant, or in genes that might benefit the bacteria were found across different lineages of *M. tuberculosis*. Dense work was developed in order to try to understand deletions/SV/duplications, and how these events can affect the pathogen. Understanding how the MTBC complex can evade and cause disease, and understanding how the genomes evolve within hosts are crucial information, however I have few concerns with the manuscript. Please evaluate the attach file containing the review.

From attached document of reviewer #2

Comments to the authors

The manuscript describes in silico evaluation of the pangenome of 109 long-read sequencing *M. tuberculosis* genomes. Several duplications involving antibiotic resistant, or in genes that might benefit the bacteria were found across different lineages of *M. tuberculosis*. Dense work was developed in order to try to understand deletions/SV/duplications, and how these events can affect the pathogen. Understanding how the MTBC complex can evade and cause disease, and understanding how the genomes evolve within hosts are crucial information, however I have few concerns with the manuscript.

Main points -

1- The literature review was not complete, and in some sentences the authors tend to describe that there are not other studies. However, there are many other pangenome studies, even with more 100 genomes.

Yes, there have been pangenome studies published previously that examine the *Mtb* pangenome using more than 100 isolates. However, nearly all of these with larger gene sets utilize short-read sequencing predominately or exclusively, which precludes accurate *de novo* assembly, especially for the many repetitive regions in the *pe/ppe* genes (PMIDs 33502304 & 35020793), which comprise 10% of the *Mtb* proteome. We intend to communicate that we host in-house a large set of long-read sequenced, reference quality, *de novo* assembled genomes to which we have applied a gene-adjacency graph-based pangenome tool. While many studies include genomes with some of these features or apply this tool, the combination of these features in a large set of genomes is something that makes our dataset of in-house genomes one of the larger datasets to fill such a perspective in pangenome analyses. We have expanded our literature search to include current pangenome studies which also applied long-read sequencing, as well as datasets that demonstrate the breadth of pangenome studies in association with clinical and phenotypic features of *M. tuberculosis* complex; this search includes the reviewer's suggested literature which we thank the reviewer for. The references can be found in supplemental data file 1, as well as cited in the section which discusses pangenome studies by focus beginning in line 112.

The beginning and end portions of the section are copied here with elipses:

“The foci of recent pangenome studies are diverse, spanning discovery of vaccine candidates, drug targets, and antibiotic resistance targets (58–64) (Supplemental Data File 1)... These studies have also reported whether the shared genetic pool among *Mycobacteria* is open and expanding or closed.”

2- Lineage categorization was not well defined. The authors mention the occurrence of *Mtb* lineages 5 and 6, but they are caused by *Mycobacterium africanum*. There is no indication that other MTBC members were included in the manuscript. Also, how the lineages were categorized, by RDs?

We thank the reviewer for bringing to our attention the lack of detail regarding lineage categorization. We agree that this necessary information should be provided, and have added the missing explanation of lineage determination to section 5.1 our Materials and Methods. Regarding lineages 5 and 6, they are legitimate lineages of the human-adapted pathogen despite being referred to as a different species, and this is the convention applied to this manuscript. To provide the necessary information which clarifies the determination of lineages for our isolates, we have added the sentence below to the document, located beginning in line 779.

“Lineages were called using TBProfiler version 4.1.1 and database version 7fc199d (154).”

3- Were any not *Mtb* genomes detected? Many times, some *Mtb* datasets include *M. bovis* by mistake, for example. That’s why I recommend RD check – described by Warren et al., 2006.

Non-*Mtb* genomes were not detected by the method used to determine lineage categorization: TB-Profiler (PMID: 36584023). This method is capable of detecting non-TB strains like *M. bovis* as well. The method used here can determine lineages comparably to the method described by the reviewer.

4- Additionally, it’s important to check for heterogeneous mutations - in case of contamination of the sample or even infection with 2 different *Mtb* strains - GATK, varscan, and others offer the option to evaluate heterogeneous calling.

We thank the reviewer for this comment, which considers the possibility of genomic heterogeneity contained in the isolates used for this pangenome analysis. The scope of this project was focused on the consensus genome, and so we did not systematically search for minor variant populations. This is an important analysis, and it is planned for a separate work by our group. Regarding consensus quality, when these genomes were assembled, all areas of the genomes passed quality check (see supplemental methods). All assemblies resolved into a single contig and did not contain pervasive heterogeneous structural variants that would indicate misassembly. The read support for all genomic regions was similar, and no areas of anomalously high sequencing coverage depth were present which would indicate multiple genomes. We are therefore confident that the consensus genomes represent singleton clinical isolates (rather than a chimeric assembly of mixed infection or contamination). Assembly quality control indicated no genomic regions with suspect quality measures that would indicate issues with assembly. This pangenome analysis does not analyze heterogeneity at the sub-consensus level or strive to detect minor subpopulations. While such an analysis is of scientific interest, it is not essential for the questions asked and answered in our investigation and is thus outside the scope of this manuscript.

5- The authors assume that the dataset used has high quality because the genomes are long read, but long read errors were not considered. And some sentences induce the reader to believe that all the other thousands of genomes available in NCBI have poor quality.

We thank the reviewers for indicating where we could better explain the rigor of our quality control steps which convince us of the quality of this data. Our aim is to indicate the potential of long-read sequenced, *de novo* assembled genomes to provide in-depth information about MTBC genomic features which might otherwise be missed due to methodological limitations. We have assuaged our language throughout the manuscript which wrongfully implies poor quality of other publicly available genomes, refocusing on the strengths of our study that can contribute to the breadth of pangenome studies available. Long read sequencing can incur a high cost per base, have misidentify single-base variants at a considerable rate, and reach low read depth if one is not careful (doi: 10.1007/978-981-13-8844-6_15, DOI: 10.1186/s13059-022-02604-2, DOI: 10.1186/s13059-020-1935-5). The points made in the sentence above exactly have been added to our discussion of long-read sequencing, which can be found in the introduction of our work, lines 107-109.

Additionally, the authors are not considering the combination of short+long sequencing to prevent base calling, or SV/deletion errors.

While short + long-read sequencing can produce good results, it is not necessary with sequencing chemistries P6C4 and later, which reliably produce reference-quality prokaryotic genome assemblies, and have been used to correct previous reference genomes (PMID: 28415976). Adding short-reads is not necessary to prevent base-calling errors, and are poor at detecting structural variations, as their length precludes confident mapping and unambiguous detection of variations that exceed their length. We and others have published instances where short-read sequencing missed important structural variants detected by long-read sequencing (PMIDs 28415976, PMC7591249, 36030622, & 36992927, among others).

6- Unique genes – long read sequencing can generate wrong calling, with miscalling small gene. The authors checked with blastn in only the dataset created in the study, or within NCBI dataset? That was not clear.

We thank the reviewer for calling attention to this detail. Regarding the concern of long read sequencing miscalling small genes, please see our response to the concern raised about line 141 in our original submission.

As for the second question, we can understand the lack of clarity on our behalf which introduces confusion. Our use of BLASTN with the unique genes was to answer two different questions. First, since we originally used Prokka to generate input GFFs for our clinical isolates but used the curated GFF of H37Rv from NCBI as input for Panaroo, we hypothesized that the reason we did not find any of these unique genes in H37Rv could be because of that pre-processing difference. Thus, we wanted to differentiate between unique genes which were just missed unannotated open reading frames in H37Rv versus genes that were putatively novel. To do this, we used Prokka to identify open reading frames in H37Rv that did not overlap in coordinates with a known gene, and aligned those sequences with BLASTN to our Panaroo-derived pangenome multi-FASTA file to identify which of H37Rv's unnamed open reading frames were the same as the putatively novel genes found in the other clinical isolates. Second, we wanted to see from which genes the putatively novel genes were derivatives. Therefore, we then took the sequences of the putatively novel genes from the pangenome multi-FASTA file and aligned them to the H37Rv-NCBI genome using BLASTN. This helped us identify that many putatively novel genes were likely mutated gene fragments (see Supplemental Data File 4).

The changed methods for added clarity are as follows in lines 838-850: "Prokka v. 1.14.6 was run on H37Rv FASTA file from NCBI (H37Rv-NC_000962.3) with the same input parameters as

the clinical isolates in order to identify which Panaroo-identified unique genes were in fact present in H37Rv as unnamed open reading frames. The open reading frames that were not assigned a gene name in the Prokka-produced GFF of H37Rv were extracted. Start and stop locations of these open reading frames were compared with the H37Rv-NCBI curated GFF file to filter out open reading frames that overlapped in nucleotide position with other annotated genes. The nucleotide sequences from the remaining unnamed open reading frames were aligned to the Panaroo-derived reference multi-FASTA file using BLASTN (162) to identify putative novel genes in our pangenome that are missed from the NCBI version of the H37Rv annotation.

We then sought to roughly characterize from which genes the putative novel genes may be derivatives as gene duplicates or gene fragments. Thus, Panaroo-derived sequences of these genes were all aligned to the NCBI H37Rv genome using BLASTN with default parameters. The best hits according to percent identity and query coverage were reported.”

Language was also updated in the corresponding results section in lines 335-339 and 358-363 to add clarity.

7- The bio projects used in the study mention 92 and 70 isolates, please clarify with a new table each isolate, adding specific information of each - SRA ID, lineage, atb resistance, and other pertinent information. Also, why these datasets were selected, against other studies that also used long read? Please clarify

For this study, we selected isolates within our in-house biobank that were sequenced on PacBio RS II using the P6C4 chemistry, and had completed *de novo* assemblies that passed initial quality control assessments. This is the reason that not all isolates from the 92 and 70 isolate studies were used. A new table has been created to include pertinent information about each isolate and has been added as Supplemental Data File 7. We have included the Biosample accession numbers in this data file, which will map to the SRA data once it is all fully uploaded.

Some details that can be better evaluated:

Line 19 – first time mentioning *Mycobacterium tuberculosis*; don't abbreviate

This has been corrected, and can be found in its corrected form in line 20, as well as directly quoted below:

“*Mycobacterium tuberculosis* (*M. tuberculosis*) complex successfully adapts to environmental pressures.”

Line 20 - that is not the first study to use many complete genomes to evaluate pan genome, the authors are indicating that other studies were made with bad quality genomes? Please provide better data for that.

Our intent was to highlight that to our knowledge, we were the first to use high quality, complete *de novo* assembled genomes from long-read sequencing to assess the pangenome specifically using a graph-based analysis method. While other studies have used long reads and/or *de novo* assembled genomes to assess the *M. tuberculosis* pangenome, this combination of considerations in conjunction with a graph-based tool to more comprehensively assess gene adjacency and structural order is a gap in pangenomes prior; our considerations could contribute to and reinforce prior work. Although the founders of Panaroo have used *M.*

tuberculosis data to showcase Panaroo as a pangenome analysis tool, they only used short read sequenced genomes which bring their own limitations (see above comments to question 5). To clarify our language, the sentence has been revised to the following in lines 22-24:

“In this study, we used 110 reference-quality, complete *de novo* assembled, long-read sequenced clinical genomes to study patterns of structural adaptation through a graph-based pangenome analysis, anticipating rarely studied mechanisms that enable enhanced clinical phenotypes.”

Line 39 - there are other old pathogens involved with human evolution, I recommend the authors to rephrase this sentence

We thank the reviewer for this comment. This sentence has been rephrased, and can be found in lines 41-42, as well as directly quoted below:

“Importance: *M. tuberculosis* complex (MTBC) has killed over a billion people in the past 200 years alone and continues to kill nearly 1.5 million annually.”

Line 45 – *de novo* should be italicized

This has been corrected, and the correction can be found in lines 47-48 as well as directly quoted below:

“In this article we have *de novo* assembled 110 clinical genomes (the largest *de novo* assembled set to date) and performed a pangenomic analysis”

76 - Would be Mtb lineages instead of complex?

Yes indeed, the deletions are correspondent to Mycobacterial human-adapted lineages. This correction has been made and can be found quoted directly below, as well as in lines 75-77.

“Many studies have identified specific deletions, termed RvD1-6 and TbD1, in the most studied *M. tuberculosis* reference strain, H37Rv, with respect to *Mycobacterium* human-adapted lineages (21,22).

80 - that is not correct, please review literature

We thank the reviewer for bringing to our attention the need for more careful review of the literature surrounding the application of long-read sequencing to pangenome studies of MTBC. We have found a series of studies focusing on specific lineages, *Mycobacterium bovis*, MTBC causative agents for extrapulmonary tuberculosis, and the impact of gene loss on diversification (PMID: 36677470, PMID: 37266022, PMID: 34241588, PMID: 38230932). We have additionally found separate instances of the application of long-read sequencing to MTBC containing investigations, including them in our citations. We have amended our statements, changing the sentences located starting with the line cited by the reviewer. The corrected language can be found in lines 101-107 and can also be found quoted below:

“Studies of a limited number of *de novo* assembled MTBC genomes revealed such genomic changes(45–48). A few pangenome studies employing long-read sequencing of different constituents of MTBC, specific lineages of *M. tuberculosis*, and of causative agents for cutaneous tuberculosis have demonstrated the benefits of long-read sequencing for characterizing genetic diversity (26–30). Application of long read sequencing to *M. tuberculosis*

for distinct purposes separate from pangenome analyses have additionally been found, showing the promise of such technology to many fields of tuberculosis research (49–52).

119 - there are thousands of mtb genomes, why the specific dataset used in the study would be better? Why not consider then, all NCBI datasets? The authors should compare with other data/genomes to suggest that only these genomes have the best quality

We thank the reviewer for this comment. These genomes in this study derive from isolates collected through collaboration and were sequenced under our group's investigation. Our study is not using a collection of publicly available data, rather our data produced in-house. The genomes are published in NCBI as they have been analyzed in other manuscripts by our group as well. The isolates were long-read sequenced and contain isolates from all human-adapted lineages, and a wide array of drug resistance patterns. Because of the quality and completeness of these isolates (motivated in page 43 of our supplemental information) and the benefits of long-read sequencing in resolving areas of the genome usually omitted from *M. tuberculosis* such as *pe/ppe* gene families (line 86-87) we posit that our isolates can contribute knowledge to the area of pangenomic analysis as applied to *Mycobacterium tuberculosis* complex. The genetic content reported in pangenomic analysis depends heavily on the accuracy of library prep, sequencing and assembly, which we have verified through our quality control checks of these genomes (page 42 of our supplementary methods). Our choice to not use public data is not because we claim all other investigators produce poor quality genomes, but by using our in-house genomes, we can be certain of the quality of these genomes from dna extraction to assembly.

We strive to limit the conflation of this statement with statements of potential limitations from previous studies. We feel our genomes pose strengths that confirm previous findings and contribute new ones, especially when previous work has shown that short-read sequencing of *M. tuberculosis* isolates contains regions of low confidence calls (<https://doi.org/10.1093/bioinformatics/btac023> ,<https://pubmed.ncbi.nlm.nih.gov/33502304/>). Language that assuages our statements and clarifies our position has been added. This language can be found quoted directly below, and can also be found in lines 83-111.

“Short-read sequencing imposes limitations that might not be overcome post-sequencing...

“However, our study contributes to past genome comparison studies (29,53–59), which have employed some *de novo* assembled genomes, to accurately describe the diversity of MTBC in these difficult-to-characterize regions.”

Line 131 - please, use comma for numbers (4,325) - nomenclature as line 479

We thank the reviewer for this comment. A check of the document has been made, and numbers over 999 have been corrected to include a comma. This document check included tables as well. Regarding the specific line mentioned in the comment above, a correction can be found in lines 146-147 and can also be found quoted directly below:

“The pangenome of 109 MTBC clinical isolates and H37Rv contained a total of 4,325 genes, reflecting the total number of genes collectively present in any of the genomes.”

141 - Careful with unique genes - specially older long read sequencing the creating of small

gene fragments was a problem, leading to the use of illumina that would work better with high gc content mycobacteria. Please check those genes if they are really unique or just miss annotation

We thank the reviewers for this comment. SMRT-sequencing chemistries prior to P6C4 indeed had high rates of sequencing errors, particularly in single base indels. P6C4 chemistry reads can sometimes have the same issue as well, to a lesser degree, when the SMRT cell is overloaded or underloaded. Upstream of our quality control on the sequencing data, we have performed a significant amount of long-read sequencing and worked with the sequencing center to limit the possibility of over/under-loading. We describe the quality control steps for assembly quality in the Supplemental Material.

Importantly, the sequencing errors with long reads are random, rather than systematic. Therefore, as sequencing coverage depth increases, so too does the consensus accuracy. We sequenced at a coverage depth high enough for this consensus accuracy to exceed QV60 (one error per million bases). This accuracy is corroborated by our long-read sequencing of biological replicates of H37Ra using the same protocol (PMID: **28415976**), which produced nearly identical assemblies (over QV60) and were able to correct several dozen sequencing errors in the previous, Sanger-sequenced H37Ra reference genome. We similarly assessed biological triplicates of H37Rv sequenced with the same chemistry and quality control pipeline, which were de novo assembled independently, with two sequences matching identically and the third having a single base insertion in the entire genome.

In contrast, Illumina has *systematic* errors associated with high GC content, and particularly struggles with unambiguously mapping short sequencing reads to the genome. This systematic error is exacerbated in the numerous regions of sequence similarity and high GC content in many *M. tuberculosis* *pe/ppa* genes, which is often the reason for their exclusion from other pangenome studies. This has been demonstrated by our group (PMID: 33502304) and others (PMID: 35020793) for *M. tuberculosis*.

Lines 141-144 - that would be better fit in discussion

We thank the reviewer for this comment, which motivates re-location of this section of text to the discussion section, where it is better suited. We agree with the reviewer that this comment should be re-located to the discussion section, and have found an appropriate section, where we discuss the prominence of lipid metabolism processes in our pangenome as also motivated by a separate analysis. The text can now be found in lines 649-659 and the section of text in addition to text preceding and succeeding it can be found copied below.

“The prominent trehalose transport pathway is differentially distinct among our isolates, and contributes to caseous granuloma catabolism processes, macrophage lysosome-phagosome fusion, IL-12 mediated modulation of IFN- γ expression which centrally facilitates canonical granuloma formation, and the development of post-primary disease due to the high lipid content. The enrichment of lipid metabolism processes in the core genome through GO analysis highlights the centrality of functions that sustain the bacteria in-host within macrophages (49) and granuloma, the immune-cell environment that encases infection in the lung. This phenomenon as supported by these two orthogonal analyses has not been specifically named in previous pangenome analyses as an important delineating factor among isolates, despite these processes contributing to the high virulence of MTBC which can vary among isolates and even lineages.”

149 - when lineages were classified? and how?

Lineages were classified during the sequencing efforts (2015-2019) using TBProfiler version 4.1.1 and database version 7fc199d (PMID: 36584023). This information has been included in the Materials and Methods section of our manuscript and can be found as stated above on lines 779-780.

154-158 – discussion.

We thank the reviewers for this comment, which suggests the re-location of text discussing the H37Rv genome's distinction from other L4 isolates. Because this text does not discuss empirically-derived results rather points of discourse surrounding results, we agree with the reviewer that this text is better suited for the discussion section. As such, the text has been relocated to the discussion section where other points regarding H37Rv are made, and can be found in lines 710-720. The section encased in text preceding and succeeding can be found copied below for the reviewer.

“Our analyses also highlighted important differences between our diverse set of clinical isolates and the H37Rv reference genome. While the proportion of putative novel genes remained consistent across clinical isolates, H37Rv contained a smaller number and proportion of these genes. These differences resulted in H37Rv clustering separate of all other isolates, including other lineage 4 isolates. Removing H37Rv from the L4 lineage-specific pangenome analysis increased the number of genes shared across all L4 isolates from 3,678 to 3,826 genes. This increase in L4 core genes suggests the genomic composition of H37Rv differed from other isolates within L4 lineage but does not solely explain the difference in diversity seen in L4 compared to other lineages. The notable difference between H37Rv and the rest of L4 may be attributed to the fact that H37Rv has become a laboratory strain for the past several decades and has undergone a separate evolutionary path. This underscores the genomic differences in H37Rv from clinical isolates acknowledged by others (29,41).

Figure 1 - here the authors include *M. africanum* (lineage 5 and 6), these two species are not *M. tuberculosis*, but are in MTBC complex. The methodology only includes *M. tuberculosis* in the sampling - please clarify if different species from MTBC were included.

We thank the reviewer for bringing this to our attention and apologize for the oversight. A check of the document has been made to clarify that isolates used include other species of *Mycobacterium tuberculosis* complex, not just *M. tuberculosis*.

Line 161 – figure 1 legend - n= should be italicized, also, keep consistent, some lines are presented with “N”, some with “n”.

This has been corrected throughout the document to align with standard convention, and we thank the reviewer for bringing this forth. Gene names are italicized and written in lower case. Protein names are not italicized and the first letter is in uppercase. The specified correction regarding figure 1 (caption located in lines 218-219 can be found below:

“Figure 1. Distribution of genes shared by lineage in the pangenome. (A) Diagram showing both the number of genes shared and unique to the lineages in the study sample ($N=110$ isolates).”

Line 174 – wouldn't be better to start results showing all the dataset, showing the lineages, antibiotic resistance and other features? Maybe at ~ line 145

We thank the reviewer for this suggestion and have included information regarding the isolates in the results. This information includes lineage designation (how many isolates belonged to what lineage) and the isolates' drug resistance classification. We have also included major GO gene sets enriched across isolates, and broad patterns among isolates in the beginning of the results section of the manuscript. The beginning snippet of this information is directly quoted below and can also be found in lines 151-156 in the clean version of the manuscript.

“This dataset included lineage 1 (L1) isolates ($N=20$), lineage 2 (L2) isolates ($N=37$), lineage 3 (L3) isolates ($N=10$), lineage 4 (L4) isolates ($N=38$), lineage 5 (L5) isolates ($N=5$), a lineage 6 (L6) isolate ($N=1$), and a lineage 7 (L7) isolate ($N=1$). There were 3 multi-drug resistant isolates (MDR), 22 mono-resistant isolates, 3 pan-susceptible isolates, 10 pre-extensively (XDR) drug resistant isolates, and 8 isolates with other drug resistance patterns. There were 11 isolates for which drug-resistance information was not available (**Table 1, Figure 1**).”

Line 209 - first time introducing the species, please use full scientific name - *Mycobacterium leprae*, ...

Species nomenclature has been corrected in response to this comment, and we thank the reviewer for highlighting this misnomer. The correction can be found in lines 266-269 of the clean version of the manuscript and is also quoted directly below:

“Orthologous genes shared across *Mycobacterium leprae*, *Mycobacterium smegmatis*, *Mycobacterium avium* and *Mycobacterium abscessus* have 6.64 times the odds of being conserved in the core, highlighting genes that are likely broadly essential to *Mycobacterium* species (FDR corrected p -value = 1.18×10^{-25} , **Table 2**).”

Line 215 – check gene vs protein nomenclature.

We thank the reviewer for this comment. We have reviewed and corrected gene/protein nomenclature at the line specified by the reviewer. We also verified that the nomenclature is correct in other sections of the paper. The instance brought forth by the reviewer has been changed, with its corrected form available in lines 275-277 and is also quoted directly below:

“Conversely, *pe/ppc* genes had significantly reduced odds of being in the core compared to the accessory genome, highlighting their variability in number across our sample (Table 2)”

Line 232-233 – reference to support the statement.

We thank the reviewer for this comment, which addresses the need for support to our statement found across lines 287-290 (copied here): “Our use of long-read sequencing allows us to

overcome these challenges. The diversity in presence and count of these genes can attenuate or enhance the aforementioned phenotypic characteristics of an isolate, making the correct identification and characterization of these gene families' constituents vastly important (101,102).”

Prior work from our group has demonstrated the advantage of applying long read sequencing to resolve issues which arise when characterizing the number and sequence of the highly recombinatorial, repetitive, and high GC rich genomic regions comprising *pe/pe_pgrs/esx* gene families' sequences (PMCID: PMC5393005). Prior work from our group (PMCID: PMC8190613) has also discovered a limitation of short-read sequencing which could compromise correct identification of *pe/pe_pgrs/esx* genes, which made up over half of identified blind spots in short-read sequenced genomes. The careful analysis of these gene families is necessary, as these canonical genes bolster a variety of functions necessary to mycobacterial function and infection. These functions were enumerated at the beginning of the paragraph and supported through references cited there. If the number and identity of these gene families cannot be correctly identified, then the proposed functions and processes of those isolates is not correctly captured, which can have consequences for the reported diversity and function of the present genes, and ultimately isolate functions and processes. To align with this, the text has been changed, and changed text can be found in lines 288-290, as well as copied above). Additionally, a citation has been added to the text to support our statement: PMID: 37876785, where identified *pe/ppe* genes created diversity among lineages, and created differences in pathogenicity, an important function and process of *Mycobacteria*. We have also added a second citation (PMID: 25727695) where the diversity of these gene families is discussed and the implications of this diversity on mycobacterial-host-interaction and host response, an area of important function.

Line 234 – and other lines, please, include Rv gene ID among the studied genes

We have updated the manuscript so that the Rv gene name is included with the first mention of each single gene in the text.

Line 236 – table S1 - please, make the sequence available.

We have provided sequences to the genes in Table S1, which is available in the new Supplemental Data File 5.

Line 275 - Would be better to combine this topic when fig 1 is first introduced

Indeed, this section is better suited closer to when higher-level patterns and themes of our pangenome are described. This section has been moved from section 2.2 to 2.1 right after our GO analysis and description of putative novel genes. This way, from a data-driven perspective, the reader is oriented to the dataset and characteristics such as lineage and drug resistance early on. The section can be found beginning in line 180

“2.1 Clustering and principal component analysis of gene presence profiles reveal putative novel genes to be significant distinguishing factors between lineages...

Line 276 – wouldn't be among lineages?

The PCA analysis and hierarchical clustering analyses would demonstrate the isolate-level differences through either combination of genes or all genes in the sample space, and this separation and distance among isolates could then be attributed to factors such as lineage, or other factors influencing samples.

Line 327 - what about other mtb genomes available in NCBI, were the genes ever described?

The only genome from NCBI included in this study was the NCBI version of the reference genome H37Rv. As Panaroo required GFF files as input, we used the NCBI curated GFF of H37Rv as input with our in-house isolate GFFS produced with Prokka. The comparison of genomic content of the clinical isolates compared to H37Rv was to raise awareness of its limitations to other researchers that depend on this version of H37Rv for reference-based assembly and annotation. Given the difference in genomic content, aligning to this genome would ultimately result in loss of annotated genes. This finding remained consistent when evaluating with a Prokka-produced GFF using the NCBI H37Rv-NC_000962.3 FASTA file. Our intent was not to query other genomes available on NCBI, but rather report on the impact of using this particular NCBI genome (i.e., H37Rv) for reference-based alignment.

Line 358 - against what dataset? the one generated herein, or against all NCBI.

In this section, we report on our alignment of the putatively novel genes against the NCBI version of the H37Rv genome. We did not use BLASTN to align to all genomes in the NCBI database, as our goal was not to identify orthologs, but to see if we can get distinct alignments to known H37Rv genes.

Line 420-421 - is there any data to support the drug resistant, or reference?

This is supported by the phenotypic drug susceptibility testing that was completed for these isolates (PMID: 24353002 and PMID: 25859997).

453 - reference to support the survival advantage.

The references to support the claim about survival advantage are a few sentences later in lines 481-485:

“We found that the *ce/A1_g-402a* variant caused the creation of a SigI_10 sigma factor motif (G[AGT][AGCT][GCA][TAG][CA], or GDNVDM in terms of IUPAC nucleotide ambiguity codes). Extra-cytoplasmic function (ECF) sigma factors like SigI are important for response to adverse environmental stressors(113,114). Creation of a SigI binding motif preceding *ce/A1* may allow rapid regulation of biofilm, thus aiding its survival.”

While the sentence in question at line 453 of the original submission was meant to introduce this line of inquiry, I can understand how it appears unsupported with its current presentation. As such, this sentence has been removed.

Line 575 - the authors included *M. africanum*

The language in this sentence has been changed to reflect the inclusion of *M. tuberculosis* complex species isolates in the analysis. The correction can be noted below, as well as on lines 588-589.

“Our pangenome analysis showcases regions of the *M. tuberculosis* complex genome that exhibit the greatest genomic diversity across lineages.”

Line 576-577 - Please explain why the picked bio projects would have better quality, compared to all the other dataset available in SRA; It’s known that the use of illumination + long read can give better genome, I recommend the authors to include those genomes to evaluate how is the quality of only long reads vs combine.

We thank the reviewer for this suggestion. Please see our response to Main Point 7 above for justification as to why we selected these genomes. In reference to adding Illumina sequencing with our long reads, please see our response to Main Point 5.

Line 583 – *in vivo* – italicize

The word referenced by the reviewer has been italicized. The corrected sentence can be found below, and additionally in lines 596-597.

“Our finding that numerous *in vitro* non-essential genes are in the core genome suggests that many non-essential core genes *in vitro* have essential roles during infection or transmission.”

Lines 588-590 - there are many other papers that the authors should consider to compare and to mention

- <https://journals.plos.org/plosone/article?id=10.1371/journal.pone.0175330>
- <https://www.frontiersin.org/journals/pharmacology/articles/10.3389/fphar.2018.01492/full>
- <https://www.mdpi.com/2079-6382/10/5/500>
- <https://www.mdpi.com/2076-2607/11/6/1495>
- <https://pubmed.ncbi.nlm.nih.gov/29259589/>
- <https://www.ncbi.nlm.nih.gov/pmc/articles/PMC9676053/>

We thank the reviewer for the additional sources that can be used to contextualize our pangenome analysis. The mentioned publications regard pangenome and comparative genomic analyses that used Illumina or other platform based short-read sequencing (PMID: 37374997, PMID: 30618776, PMID: 29259589) with either reference based (PMID:

28394899, PMID 33924811) or *de novo* assembly (29259589) to produce complete genomes. These genomes of good quality available through public repositories explore pangenomes of a variety of *M. tuberculosis* or *M. tuberculosis* complex isolates and their association to virulence of human adapted strains compared to BCG (PMID 33924811), distinguishing factors of a Lineage 1 Manila strain (PMID: 28394899) compared to H37Rv and a Beijing strain isolate, clinical features such as isolate site (pulmonary versus extrapulmonary infection; PMID 37374997), extended-spectrum extensive drug resistance (PMID 30618776), and the variation in pseudogenization among MTBC strains (PMID 36250787).

These analyses provided an understanding of variants and evolutionary divergences among MTBC that contributed to higher rates of transmission, increased success of host tissue colonization, and targets for therapeutics to aid in the eradication of tuberculosis infection similar to our study, though the included genomes might vary in strain (*M. bovis* and *M. canettii*, for example) or lineage. We have come to understand that these studies contain complete genomes or apply *de novo* assembly similar to our study, and include a similar or larger number of genomes. Our dataset contributes a graph-based pangenome approach that was also employed by a study, though we consider our application of this method to our in-house long-read sequenced, *de novo* assembled, complete, reference quality genomes as an additional contribution to pangenome studies of MTBC. We have produced additional insight into aggregates of genomic content (multiple-gene adjacency variants) as an avenue of investigation towards contributors of virulence and host colonization; this has not been considered in these studies. We additionally discuss the count of genes from important gene families (*ppe/pe/esx*) found in our in-house genomes, genetic content not aligning with previous known genetic sequences, and duplications of genetic content contributing to drug resistance. We hope to continue to explore how MTBC genomes sequenced and assembled with our considerations contribute knowledge to this expanding area of research (MTBC pangenome studies). Additionally, we are excited by the results that we have found in our genomes also seen in the publications brought forth by the reviewer (the importance of lipid metabolism pathways in host tissue colonization and the divergence from H37Rv strain implicating its own evolutionary path), which add external confirmation of our findings.

To better summarize prior work and contextualize our findings, these publications have been added to our literature review summary, among other pangenome analyses that demonstrate the breadth of current pangenome analyses. The section with the included citations can be found in lines 81-124 (the start and end are copied below to bookend the section:

“A few studies have employed long-read sequencing, and (26–30), many have used short-read sequencing to assess genomic diversity of *M. tuberculosis* through the construction of its pangenome... While pangenome studies on the shared genetic pool between human-associated and animal-associated pangenomes are important, the inclusion of non-tuberculous *Mycobacteria* strains has reported a smaller and more highly conserved core genome (since conservation is defined as genomic features shared among these varied-host strains).”

Line 861 – explain what isolate is that - SEA11278; add table with all SRA ids, and different info.

A table with pertinent isolate information including biosample and bioproject IDs has been added as Supplemental Data File 7. This isolate in particular is a monoresistant Lineage 7 isolate, which our hierarchical clustering analysis demonstrated as having the furthest distance to other isolates. This isolate was considered as an outgroup to the other isolates and could serve as an anchoring isolate to base the backbone of the adjacency network. This is because any differences among isolates would exist within the range of variability between this isolate and H37Rv. The biosample and bioproject identifiers have been added to further identify this isolate. Amended language has been added to line 885-888 and can also be found quoted below:

“...isolate SEA11278, a monoresistant lineage 7 isolate (biosample SAMN12325265 bioproject PRJNA555636) that is most dissimilar (demonstrated through hierarchical clustering analysis; **Figure 1**) to the other isolates present in this pangenome, then circularized for downstream network analysis.”

Re: mSystems00499-24R1 (Interred mechanisms of resistance, host immune evasion, and an incomplete perspective on essentiality revealed through network-connectivity analysis of *M. tuberculosis* Complex graph pangenome)

Dear Dr. Faramarz Valafar:

Thank you for revising the manuscript. The article has been re-reviewed by one of the initial reviewers and there are a few additional comments that have been brought up that still remain. I agree with the reviewer comments and believe that by addressing these, the manuscript would be substantially improved. Please find reviewer comments below for your consideration.

Revision Guidelines

Sincerely,
Zackery Bulman
Editor
mSystems

Reviewer #1 (Comments for the Author):

Several concerns remain in the revised manuscript.

1. The authors report that core genes account for over 85% of the total genes in mtb, while previous studies indicate that only

10-15% of genes are essential. Given these figures, it is evident that the majority of core genes are non-essential, which is not surprising. I am unclear how this finding challenges the current understanding of gene essentiality.

2. The definition of "novel genes" requires further clarification. Initially, I interpreted this to mean gene sequences not found in reference genomes. It would be more logical to separate "novel genes" into two categories: (1) newly annotated genes (sequences present in the reference genome but previously unannotated as genes) and (2) true novel genes (sequences absent from reference genomes). Another concern is that the study relies solely on Prokka for gene prediction. Results for "novel genes" could vary if a different gene prediction tool or parameters were used.

3. It is crucial to identify which novel genes are associated with specific lineages and to determine whether they can serve as markers for lineage differentiation. Offering examples of these sequences would be valuable for the field.

4. The manuscript assumes that all 110 genomes are of high quality due to long-read sequencing and de novo assembly. However, the authors also mention the possibility of sequencing errors (line 465). To enhance the study's rigor, additional validation procedures-especially for novel or fragmented genes-should be implemented. Methods such as Sanger sequencing for selected genes or next-generation sequencing on a subset of genomes could help confirm the findings.

Subject: mSystems00499-24R1 Decision Letter
Prompts in black text, Answers in blue

Reviewer #1 (Comments for the Author):

1. The authors report that core genes account for over 85% of the total genes in *mtb*, while previous studies indicate that only 10-15% of genes are essential. Given these figures, it is evident that the majority of core genes are non-essential, which is not surprising. I am unclear how this finding challenges the current understanding of gene essentiality.

We agree that this finding doesn't challenge the understanding of gene essentiality by itself, but we believe that this core gene breakdown conflicts with how the concept of gene essentiality is currently framed. Literature shows that over time, genetic content that does not hold value is often slowly removed from a species' genome (PMID: 25461580, PMID: 15933207, PMID: 20700439, PMID: 36250787). While one can certainly argue that genes could be useful and have value but not be essential to basic cellular function and survival, our results suggest that the number of essential genes may currently be underestimated due to the fact that gene essentiality studies are largely determined through *in vitro* testing (PMIDs: 27407107, 25636614, 12657046, 35004921, 31239393, 21980284, 23028335, 24103077, 24315099, 23028335, 26438867, 26067605, 27936238, 28893793, 28504669, 30318462, 31363023, 14569030, 15928073, 36945430, 35112666, 16868085). This is unsurprising, as *M. tuberculosis* grown *in vitro* is removed entirely from many selective pressures levied in the environment that *M. tuberculosis* exists in naturally. While *in vitro* models of gene essentiality can capture genes necessary for basic cellular function and metabolism, they cannot capture the full extent of essentiality of genes in their natural context. This is supported by our assessment of the *in vitro* non-essential core genes, 38% of which held functions that are known to be vital for *in vivo* survival in humans. Moreover, core genes were overrepresented to a greater degree among broadly conserved Mycobacterial orthologs (odds ratio 6.64) than among *in vitro* essential genes in any of the four *in vitro* essentiality studies (odds ratios ranging from 2.19 to 4.41). Therefore, we pose that gene essentiality, as defined by *in vitro* studies, likely does not adequately reflect what is necessary for survival in the host. However, we do agree with the reviewer that this finding is overemphasized in our results (thanking the reviewer for bringing this to our attention), and thus has been attenuated in the manuscript, specifically removing language in the title, and altering the abstract (lines 26-28) and the discussion (as seen below at lines 582-588):

“Our findings also spotlight the need for more research into gene essentiality, as defined by H37Rv genes required for *in vitro* survival. Our finding that numerous *in vitro* non-essential genes are in the core genome, 38% of which provide functions known to be important for *in vivo* survival, suggests that the current definition of essentiality may underestimate and miss genes that have essential roles during infection or transmission within host. More research is needed to clarify which genes are essential within these important *in vivo* contexts in humans, including genes that are not present in H37Rv.”

2. The definition of "novel genes" requires further clarification. Initially, I interpreted this to mean gene sequences not found in reference genomes. It would be more logical to separate "novel genes" into two categories: (1) newly annotated genes (sequences present in the reference genome but previously unannotated as genes) and (2) true novel genes (sequences absent from reference genomes). Another concern is that the study relies solely on Prokka for gene prediction. Results for "novel genes" could vary if a different gene prediction tool or parameters were used.

We defined putative novel genes as *ab initio* predicted genes that have not been previously annotated in the H37Rv reference genome. This definition does not preclude open reading frames that are present but not annotated in H37Rv, although we found most of these putative novel genes are not present in H37Rv (216/290, 74.5% absent from H37Rv). We agree with the reviewer that differentiating between newly annotated genes versus truly novel (i.e., sequences absent from the reference genome) would be beneficial. Accordingly, we have added the list of these genes to our Supplemental Data File 6, and have added language to differentiate the two gene categories throughout our results at lines 337-339, 342-345, 358-360, and 374-376.

Regarding the concern about relying on Prokka for gene prediction, we agree with the reviewer that different tools or parameters may affect gene boundaries. However, inclusion of different annotation tools would necessitate a comparison of tools and a validation of each tool which has already been peer-reviewed; this would distract from the primary scope and focus of this paper: a presentation of the *M. tuberculosis* complex pangenome using a graph-based pangenome analysis tool. We have chosen not an obscure method or one that is underutilized for this work. Additionally, when Prokka was validated against other tools, it produced higher quality annotations than other programs against a reference genome, even with the inclusion of draft genomes and has maintained a place in the field (PMID: 24642063). We do recognize our sole use of Prokka as a limitation, however, and agree this limitation should be evident to readers who are considering our results or looking for use cases of this popular annotation tool for their own work. To this end, we have made this limitation more clearly stated in our discussion at lines 732-735:

“Additionally, Panaroo’s gene reporting depends on the annotated inputs from Prokka to construct the pangenome. Using different annotation tools or parameters may produce different gene boundary calls, and thus may slightly vary in the number of genes identified, especially of the putative novel genes and gene fragments.”

3. It is crucial to identify which novel genes are associated with specific lineages and to determine whether they can serve as markers for lineage differentiation. Offering examples of these sequences would be valuable for the field.

We agree with the reviewer that assessing if any of these novel genes are associated with lineage would add value to the scientific community. We thank the reviewer for this suggestion, and we assessed if any putative novel genes were associated with a specific lineage. Putative

novel genes present exclusively within a single lineage have been added to Supplemental Data File 6, as well as the sequences of the genes that were present in all isolates of a given lineage. Descriptions of these have been added to the results section in a new paragraph at lines 387-393:

“Many putative novel genes were found in only one lineage (82/290, 28.3%). Only one putative novel gene was seen in all L3 isolates, one gene was in all L4 isolates (except for H37Rv), two genes were in all L5 isolates, and three genes were only seen in our L6 isolate (Supplemental Data File 6). The putative novel genes found only in L5 and L6 matched an annotated gene present in H37Rv with high percent identity (>99%) but low coverage (<25%), including a putative novel gene with high percent identity to a portion of IS6110 at a locus in the pangenome unique to all L5 isolates. More research with expanded data sets is needed to confirm the use of these regions as potential lineage markers.”

4. The manuscript assumes that all 110 genomes are of high quality due to long-read sequencing and de novo assembly. However, the authors also mention the possibility of sequencing errors (line 465). To enhance the study's rigor, additional validation procedures- especially for novel or fragmented genes-should be implemented. Methods such as Sanger sequencing for selected genes or next-generation sequencing on a subset of genomes could help confirm the findings.

We thank the reviewer for this comment. Our statement of limitations regarding long-read sequences contextualizes the use of this technology with where the field stands and reminds the reader of the importance of gauging the sequencing technology used. We do not feel that this careful contextualization of the sequencing technologies' limitations works against the rigor of the study. We are simply acknowledging that despite its very low consensus error rate (QV60), it is possible that a handful of sequencing errors may be present in some isolates.

Research from multiple groups has unequivocally demonstrated that PacBio P6C4 long-read sequencing data generated can reliably be assembled into reference quality bacterial genomes over QV60 (less than one-in-a-million consensus error rate) for *M. tuberculosis* (e.g., PMID: 28415976, PMID: 33107429, PMID: 37876785) and bacteria generally (e.g., PMID: 31129166). Several of these studies corrected Sanger-based assemblies with PacBio P6C4 sequencing data. Moreover, other sequencing technologies such as Illumina sequencing have known pervasive issues and biases in mapping and cannot be assembled into accurate genomes due to insufficient read length, particularly in *M. tuberculosis* (PMID: 35020793, PMID: 33502304).

Re: mSystems00499-24R2 (Interred mechanisms of resistance and host immune evasion revealed through network-connectivity analysis of M. tuberculosis Complex graph pangenome)

Dear Dr. Faramarz Valafar:

Thank you for resubmitting your article to mSystems. Your manuscript has been accepted, and I am forwarding it to the ASM production staff for publication. Your paper will first be checked to make sure all elements meet the technical requirements. ASM staff will contact you if anything needs to be revised before copyediting and production can begin. Otherwise, you will be notified when your proofs are ready to be viewed.

Sincerely,

Zackery Bulman
Editor
mSystems